# EFFICIENT BACKDOOR ATTACKS FOR DEEP NEURAL NETWORKS IN REAL-WORLD SCENARIOS

**Ziqiang Li**[1,*], **Hong Sun**[1,*], **Pengfei Xia**[1], **Heng Li**[2], **Beihao Xia**[2], **Yi Wu**[1], **Bin Li**[1,†]
[1]Big Data and Decision Lab, University of Science and Technology of China.
[2]Huazhong University of Science and Technology
{iceli,hsun777,xpengfei}@mail.ustc.edu.cn
{liheng,xbh_hust}@hust.edu.cn
wuyi2021@mail.ustc.edu.cn, binli@ustc.edu.cn

## ABSTRACT

Recent deep neural networks (DNNs) have came to rely on vast amounts of training data, providing an opportunity for malicious attackers to exploit and contaminate the data to carry out backdoor attacks. However, existing backdoor attack methods make unrealistic assumptions, assuming that all training data comes from a single source and that attackers have full access to the training data. In this paper, we introduce a more realistic attack scenario where victims collect data from multiple sources, and attackers cannot access the complete training data. We refer to this scenario as **data-constrained backdoor attacks**. In such cases, previous attack methods suffer from severe efficiency degradation due to the **entanglement** between benign and poisoning features during the backdoor injection process. To tackle this problem, we introduce three CLIP-based technologies from two distinct streams: *Clean Feature Suppression* and *Poisoning Feature Augmentation* The results demonstrate remarkable improvements, with some settings achieving over **100%** improvement compared to existing attacks in data-constrained scenarios.

## 1 INTRODUCTION

Deep neural networks (DNNs) are powerful ML algorithms inspired by the human brain, used in various applications such as image recognition He et al. (2016), natural language processing Liu et al. (2023), image generation Li et al. (2023b; 2022c;b); Wu et al. (2023), and trajectory prediction Wong et al. (2022); Xia et al. (2022a). DNN effectiveness depends on training data quantity/quality. For example, Stable Diffusion (983M parameters) Rombach et al. (2022) excels in image generation due to pre-training on 5B image-text pairs. As the demand for data continues to rise, many users and businesses resort to third-party sources or online collections as a convenient means of acquiring the necessary data. However, recent studies Pan et al. (2022); Li et al. (2021b); Yang et al. (2021) have demonstrated that such practices can be maliciously exploited by attackers to contaminate the training data, significantly impairing the functionality and reliability of trained models.

The growing adoption of neural networks across different domains has made them an attractive target for malicious attacks. One particular attack technique gaining attention is the *backdoor attack* Goldblum et al. (2022); Nguyen & Tran (2020); Xia et al. (2022c). In backdoor attacks, a neural network is deliberately injected with a hidden trigger by introducing a small number of poisoning samples into the benign training set during the training. Once the model is deployed, the attacker can activate the backdoor by providing specific inputs containing the hidden trigger, causing the model to produce incorrect results. Backdoor attacks continue to present a significant and pervasive threat across multiple sectors, including image classification Gu et al. (2019), natural language processing Pan et al. (2022); Zeng et al. (2023), and malware detection Li et al. (2021a). In this paper, we focus on the widely studied field of image classification.

It's worth noting that previous backdoor attacks rely on a potentially overly broad assumption. They assume that all training data comes from a single source, and the collected source has been poisoned

---

[*]The first two authors contributed equally to this paper.
[†]Corresponding Author: Bin Li.

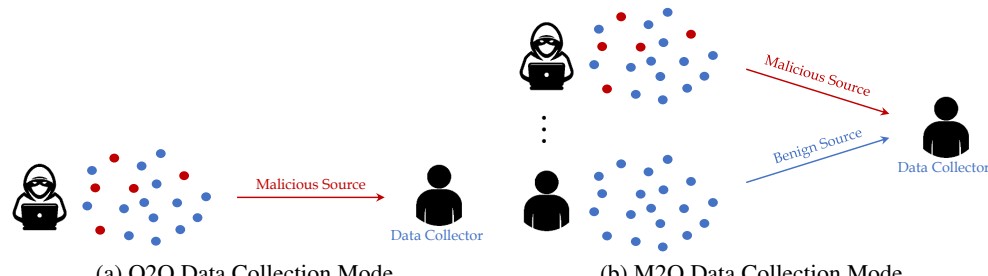

(a) O2O Data Collection Mode        (b) M2O Data Collection Mode

Figure 1: One-to-one (O2O) and many-to-one (M2O) data collection modes. M2O mode is more in line with practical scenarios where data collectors collect data from multiple sources. In this mode, the attacker cannot have all the data available to the victims.

by attacker (as shown in O2O data collection mode in Fig. 1). This assumption grants attackers full access to the entire training dataset, making it easy to poison. However, it doesn't accurately represent real-world attack scenarios. Consider a scenario where victims have a limited sample private dataset. To compensate, they may augment it by collecting additional data from various online sources (referred to as the public dataset) and combine it with their private data for training, as depicted in the M2O data collection mode in Fig. 1. In this case, some of the sources may be secretly poisoned by attackers. Attackers cannot access the private dataset and can only manipulate a portion of the public dataset for poisoning. Consequently, a discrepancy arises between the distribution of the poisoned data and the training data, deviating from previous poisoning attack pipeline.

In this paper, we address a more realistic backdoor attack scenario called **data-constrained backdoor attacks**, where the attackers do not have access to the entire training set. To be more precise, we classify data-constrained backdoor attacks into three types based on different types of data sources: number-constrained backdoor attacks, class-constrained backdoor attacks, and domain-constrained backdoor attacks. Upon investigation, we have discovered that existing attack methods exhibit significant performance degradation when dealing with these data-constrained backdoor attacks. We propose that the **entanglement** between benign and poisoning features is a crucial factor contributing to this phenomenon. Entanglement refers to the neural networks utilizing both benign and poisoning features to make decisions for poisoning samples. However, this approach is not efficient for backdoor attacks. Ideally, an efficient backdoor attack should solely rely on the poison feature generated by the trigger to make decisions, irrespective of how the benign feature is expressed. To enhance the efficiency of poisoning attacks in data-constrained backdoor scenarios, we introduce two streams: *Clean Feature Suppression* and *Poisoning Feature Augmentation* to reduce the influence of clean features and amplify the expression of poisoning features, respectively. To achieve these goals, we propose three techniques utilizing the CLIP Radford et al. (2021).

Our main contributions are summarized as follows. **i)** We present a novel and contemporary backdoor attack scenario called data-constrained backdoor attacks, which assumes that attackers lack access to the entire training data, making it a versatile and practical attack with broad applicability. **ii)** Through a systematic analysis of previous attack methods, we identify the entanglement between poisoning and benign features as the primary contributing factor to their performance degradation. **iii)** To address this issue, we introduce the pre-trained CLIP model into the field of backdoor attacks for the first time. We propose three innovative technologies: CLIP-CFE, CLIP-UAP, and CLIP-CFA. Extensive evaluations conducted on 3 datasets and 3 target models, and over **15** different settings demonstrate the significant superiority of our proposed CLIP-UAP and CLIP-CFA over existing backdoor attacks. Furthermore, CLIP-CFE complements existing attack methods and can be seamlessly integrated with them, resulting in further efficiency improvements.

## 2   BACKGROUND AND RELATED WORK

Here, we provide a concise overview of the typical pipeline for backdoor attacks on neural networks. Sec. A.1 provides a comprehensive exploration of related work on backdoor attacks and CLIP.

## 2.1 BACKDOOR ATTACKS ON NEURAL NETWORKS

Backdoor attacks aim to introduce hidden triggers into DNNs, allowing the attacked models to behave correctly on clean samples while exhibiting malicious behavior when triggered by specific inputs. These attacks can occur at various stages of Artificial Intelligence (AI) system development Gao et al. (2020). The surface of backdoor attacks has been systematically categorized into six groups: code-based Bagdasaryan & Shmatikov (2021), outsourcing, pretrained model-based Wang et al. (2020); Ge et al. (2021), poisoning-based Liao et al. (2018), collaborative learning-based Nguyen et al. (2020), and post-deployment attacks Rakin et al. (2020). Among these categories, poisoning-based backdoor attacks, which involve introducing a backdoor trigger during the training process by mixing a few poisoning samples, are the most straightforward and commonly used method. This study focuses on addressing concerns related to poisoning-based backdoor attacks.

## 2.2 GENERAL PIPELINE OF BACKDOOR ATTACKS

Consider a learning model $f(\cdot; \Theta) : X \to Y$, where $\Theta$ represents the model's parameters and $X(Y)$ denotes the input (output) space, with given dataset $\mathcal{D} \subset X \times Y$. Backdoor attacks typically involve three essential steps: *poisoning set generation*, *backdoor injection*, and *backdoor activation*.

**Poisoning set generation.** In this step, attackers employ a pre-defined poison generator $\mathcal{T}(x, t)$ to introduce a trigger $t$ into a clean sample $x$. Specifically, they select a subset $\mathcal{P}' = \{(x_i, y_i)|i = 1, \cdots, P\}$ from the clean training set $\mathcal{D} = \{(x_i, y_i)|i = 1, \cdots, N\}$ ($\mathcal{P}' \subset \mathcal{D}$, and $P \ll N$) and result in the corresponding poisoning set $\mathcal{P} = \{(x_i', k)|x_i' = \mathcal{T}(x_i, t), (x_i, y_i) \in \mathcal{P}', i = 1, \cdots, P\}$. Here, $y_i$ and $k$ represent the true label and the attack-target label of the clean sample $x_i$ and the poisoning sample $x_i'$, respectively.

**Backdoor injection.** In this step, The attackers mix the poisoning set $\mathcal{P}$ into the clean training set $\mathcal{D}$ and release the new dataset. The victims download the poisoning dataset and use it to train their own DNN models Gu et al. (2019):

$$\min_{\Theta} \quad \frac{1}{N} \sum_{(x,y) \in \mathcal{D}} L\left(f(x; \Theta), y\right) + \frac{1}{P} \sum_{(x',k) \in \mathcal{P}} L\left(f(x'; \Theta), k\right), \tag{1}$$

where $L$ is the classification loss such as the commonly used cross entropy loss. In this case, backdoor injection into DNNs has been completed silently.

**Backdoor activation.** In this step, the victims deploy their compromised DNN models on model-sharing platforms and model-selling platforms. The compromised model behaves normally when presented with benign inputs, but attackers can manipulate its predictions to align with their malicious objectives by providing specific samples containing pre-defined triggers.

## 3 DATA-CONSTRAINED BACKDOOR ATTACKS

We first show the considered pipeline of data-constrained backdoor attacks, and than illustrate the performance degradation of previous methods on proposed pipeline. Finally, we attribute the degradation to the entanglement between the benign and poisoning features during the poisoning injection.

## 3.1 PRELIMINARIES

Previous methods Chen et al. (2017); Liu et al. (2017); Li et al. (2022a); Nguyen & Tran (2020) have commonly followed the attack pipeline outlined in Sec. 2.2. However, this widely adopted pipeline relies on an overly loose assumption that all training data is collected from a single source and that the attacker has access to the entire training data, which is often not the case in real-world attack scenarios. In this paper, we focus on a more realistic scenario: *Data-constrained Backdoor Attacks*.

**Pipeline of data-constrained backdoor attacks.** The proposed pipeline also consists of three steps: *poisoning set generation*, *backdoor injection*, and *backdoor activation*. The backdoor injection and activation steps remain unchanged from the previous attack pipeline. However, in the poisoning set generation step, data-constrained attacks only assume access to a clean training set $\mathcal{D}' = \{(x_i, y_i)|i = 1, \cdots, N'\}$, which follows a different data distribution from $\mathcal{D}$. To address this,

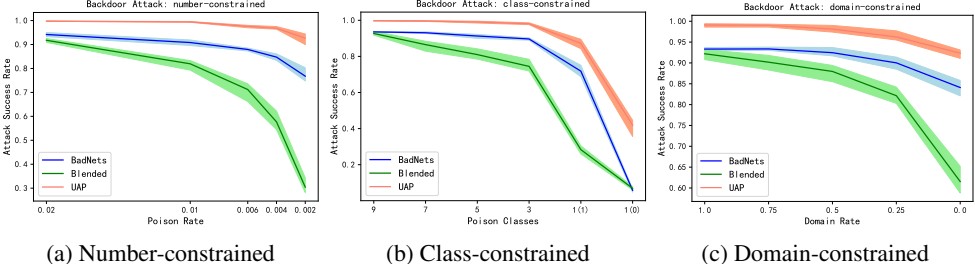

(a) Number-constrained      (b) Class-constrained      (c) Domain-constrained

Figure 2: Attack success rate (ASR) in the different data-constrained backdoor attack. The experiment is repeated 5 times, and the solid lines represent the mean results. (a): The abscissa is the number ($P$) of samples in the poisoning set $\mathcal{P}'$. (b): The experiments are conducted with triggers BadNets, Blended, and UAP, with a poisoning rate of $2\%$ ($P = 1000$) for each. The x-axis represents the number of classes ($|Y'|$) in the poisoning set $\mathcal{P}'$. Specifically, '1 (1)' and '1 (0)' denote dirty-label single-class ($Y' = \{c\}, c \neq k$) and clean-label single-class ($Y' = \{k\}$), respectively. (c): The poisoning rates of experiments with trigger BadNets, Blended, and UAP are $2\%$ ($P = 1000$), $2\%$ ($P = 1000$), and $1\%$ ($P = 500$), respectively. The abscissa is the domain rate that represents the proportion of poisoning sets sampled from $\mathcal{D} \setminus \mathcal{D}'$ and $\mathcal{D}'$.

the attacker randomly selects a subset $\mathcal{P}' = \{(x_i, y_i)|i = 1, \cdots, P\}$ from the accessible dataset $\mathcal{D}'$, and creates the corresponding poisoning set $\mathcal{P} = \{(x'_i, k)|x'_i = \mathcal{T}(x_i, t), (x_i, y_i) \in \mathcal{P}', i = 1, \cdots, P\}$. Additionally, based on the different constraints imposed by the accessible training set $\mathcal{D}'$, data-constrained backdoor attacks are further categorized into three types: *Number-constrained Backdoor Attacks*, *Class-constrained Backdoor Attacks*, and *Domain-constrained Backdoor Attacks*. The details pipeline can be found in Sec. A.2.3, Sec A.2.4, and Sec. A.2.5, respectively.

**Empirical results.** Noticing that the experimental setting of this experiment can be found in Sec. A.2.2. Fig. 2 (a), (b), and (c) illustrate the attack success rate on number-constrained, class-constrained, and domain-constrained backdoor attacks, respectively. The results demonstrate that ASR experiences a significant decrease as the number of poisoning samples ($P$), the number of class ($C'$), or domain rate (the proportion of poisoning sets sampled from $\mathcal{D} \setminus \mathcal{D}'$ and $\mathcal{D}'$) in the poisoning set decreases, particularly for Blended backdoor attacks. It is worth noting that Universal Adversarial Perturbations (UAP) achieves relatively favorable results even with a low poisoning rate. This can be attributed to the utilization of a proxy model that is pre-trained on the entire training set ($\mathcal{D}$). However, UAP is not accessible in our settings, and we present the results for UAP to effectively demonstrate the performance degradation even when a pre-trained proxy model is available.

## 3.2 ENTANGLEMENT BETWEEN BENIGN AND POISONING FEATURES

In Sec A.3, we present three observations pertaining to data-constrained backdoor attacks, serving as compelling evidence for the presence of a significant interdependence between benign and poisoning features during the backdoor injection. This intricate entanglement is identified as the primary factor responsible for the inadequacies exhibited by current attack methodologies within data-constrained scenarios. Our study is a pioneering exploration of feature entanglement in the context of backdoor attacks, yielding fresh and valuable insights into the realm of backdoor learning. Ideally, one would anticipate backdoor models to exclusively rely on poisoning features when confronted with poisoning samples, as this would be the most efficient strategy for executing successful backdoor attacks. However, neural networks tend to be greedy and utilize all features for decision-making Li et al. (2023c), leading to activation of both poisoning and benign features during backdoor injection. This results in reduced poisoning efficiency when there is a difference in benign features between the backdoor injection and activation phases, as evidenced in data-constrained backdoor attacks.

## 4 CLIP-GUIDED BACKDOOR ATTACKS METHOD

We present our approach, which consists of two components: **Clean Feature Suppression** and **Poisoning Feature Augmentation**. These components are independent of each other and can be seamlessly combined. The threat model considered in our study has been introduced in Sec. A.4

## 4.1 CLEAN FEATURE SUPPRESSION

As described in Sec. 3, the effectiveness of data-constrained backdoor attacks is hindered due to the entanglement of benign and poisoning features during the backdoor injection phase. To address this challenge, we propose a solution called "clean feature suppression" in this section. The primary objective of this approach is to minimize the impact of benign features on the decision-making process, thus amplifying the significance of poisoning features.

### 4.1.1 CLIP-BASED CLEAN FEATURE ERASING

To achieve clean feature suppression, we can employ a feature extractor pre-trained on the entire training set (As shown in Sec. **Clean Feature Erasing Noise**). However, since our data-constrained backdoor attacks lack access to the complete training set, an alternative solution is required. Recent studies have shown that pre-trained CLIP Radford et al. (2021) generates consistent and robust semantic representations across a wide range of (image, text) pairs, enabling impressive zero-shot classification performance comparable to supervised learning accuracy on challenging datasets like ImageNet (As shown in A.5). Hence, we can utilize the pre-trained general model CLIP, which replaces the feature extractor trained on the entire training set, allowing us to achieve clean feature suppression (As shown in **CLIP for Clean Feature Erasing**).

**Clean Feature Erasing Noise.** The technique of clean feature suppression aims to eliminate the clean information present in images by introducing optimized noise, denoted as $\delta$, which helps modify the input image to resemble the unbiased class. In accordance with the data-constrained backdoor attack pipeline outlined in Sec. 3, we assume that the chosen clean training dataset for generating the poisoning set consists of $P$ clean examples, denoted as $\mathcal{P}' \subset X \times Y$ (where $\mathcal{P}' = \{(x_i, y_i)|i = 1, \cdots, P\}$). Here, $x_i \in X$ represents the inputs, $y_i \in Y = \{1, 2, \cdots, C\}$ represents the labels, and $C$ denotes the total number of classes. We refer to the modified version as $\mathcal{P}_e = \{(x_{e,i}, y_i)|i = 1, \cdots, P\}$, where $x_{e,i} = x_i + \delta_i$ represents the erased version of the training example $x_i \in \mathcal{P}'$. The term $\delta_i \in \Delta$ denotes the "invisible" noise applied to achieve the erasing effect. The noise $\delta_i$ is subject to the constraint $||\delta_i||_p \leq \epsilon$, where $|| \cdot ||_p$ represents the $L_p$ norm, and $\epsilon$ is set to a small value to ensure the stealthiness of the backdoor attacks. Our objective in erasing the clean features is to ensure that the pre-trained feature extractor does not extract any meaningful information from the given images $x$. This is achieved by introducing customized and imperceptible noise, denoted as $\delta_i$. To be more specific, for a clean example $x_i$, we propose to generate the noise $\delta_i$ that erases the features by solving the following optimization problem:

$$\delta_i = \arg\min_{\delta_i} L(f'(x_i + \delta_i), y_m) \quad \text{s.t.} \quad ||\delta_i||_p \leq \epsilon, \tag{2}$$

where $L$ represents the mean squared error (MSE) loss, defined as $L(a, b) = ||a - b||^2$. The function $f'(\cdot)$ corresponds to the pre-trained feature extractor employed for noise generation. Additionally, $y_m$ denotes the unbiased label for the classification task, which is defined as $y_m = [\frac{1}{C}, \frac{1}{C}, \cdots, \frac{1}{C}]$, where $C$ signifies the total number of classes. While this vanilla method proves effective in erasing clean features, it requires a proxy feature extractor that has been pre-trained on the entire training set. This approach is not suitable for our data-restricted backdoor attacks.

**CLIP for Clean Feature Erasing (CLIP-CFE).** Taking inspiration from CLIP's approach to zero-shot classification (Sec. A.5), we leverage a general CLIP model to optimize the feature erasing noise. This allows us to relax the need for a proxy feature extractor pre-trained on the entire training set. We consider $C$ prompts, "a photo of a $c_i$," corresponding to different classes $c_i$ in the dataset, where $i = 1, \cdots, C$. The CLIP-based feature erasing noise, denoted as $\delta_i$, is proposed for the input $x_i$ by solving the following optimization problem:

$$\delta_i = \arg\min_{\delta_i} L(f_{CLIP}(x_i + \delta_i, \mathbb{P}), y_m) \quad \text{s.t.} \quad ||\delta_i||_p \leq \epsilon, \tag{3}$$

where $L$ represents the mean squared error (MSE) loss, $y_m$ denotes the unbiased label for the classification task defined as $y_m = [\frac{1}{C}, \frac{1}{C}, \cdots, \frac{1}{C}]$, $\mathbb{P}$ represents the set of prompts corresponding to different classes in the dataset, and $f_{CLIP}$ denotes the CLIP-based model used to obtain the label of the input image. Specifically,

$$\mathbb{P} = \{p_1, p_2, \cdots, p_C\} = \{\text{"a photo of a } c_i\text{"}|i = 1, 2, \cdots, C\}, \tag{4}$$

$$f_{CLIP}(x_i + \delta_i, \mathbb{P}) = \left[ \frac{\langle \hat{\mathcal{E}}_i(x_i + \delta_i), \hat{\mathcal{E}}_t(p_1) \rangle}{\sum_{i=1}^{C} \langle \hat{\mathcal{E}}_i(x_i + \delta_i), \hat{\mathcal{E}}_t(p_i) \rangle}, \cdots, \frac{\langle \hat{\mathcal{E}}_i(x_i + \delta_i), \hat{\mathcal{E}}_t(p_C) \rangle}{\sum_{i=1}^{C} \langle \hat{\mathcal{E}}_i(x_i + \delta_i), \hat{\mathcal{E}}_t(p_i) \rangle} \right]. \quad (5)$$

To solve the constrained minimization problem illustrated in Eq. 3, we utilize the first-order optimization method known as Projected Gradient Descent (PGD) Madry et al. (2017). The PGD method enables us to find a solution by iteratively updating the noise as follows:

$$\delta_i^{t+1} = \prod_\epsilon \left( \delta_i^t - \alpha \cdot \text{sign}(\nabla_\delta L(f_{CLIP}(x_i + \delta_i^t, \mathbb{P}), y_m)) \right), \quad (6)$$

where $t$ represents the current perturbation step, with a total of $T = 50$ steps. $\nabla_\delta L(f_{CLIP}(x_i + \delta_i^t, \mathbb{P}), y_m)$ denotes the gradient of the loss with respect to the input. The projection function $\prod$ is applied to restrict the noise $\delta$ within the $\epsilon$-ball (with $\epsilon = 8/255$ in our paper) around the original example $x$, ensuring it does not exceed this boundary. The step size $\alpha$ determines the magnitude of the noise update at each iteration. The resulting erasing examples are then obtained as follows:

$$\mathcal{P}_e = \{(x_{e,i}, y_i) | i = 1, \cdots, P\}, \quad \text{where} \quad x_{e,i} = x_i + \delta_i^T. \quad (7)$$

### 4.2 POISONING FEATURE AUGMENTATION

In addition to eradicating clean features in images to tackle the entanglement between benign and poisoning features, enhancing the expression of poisoning features is another effective approach. In this section, we present two parallel triggers aimed at augmenting the poisoning features: CLIP-based Contrastive Feature Augmentation (4.2.1) and CLIP-based Universal Adversarial Perturbations (A.6).

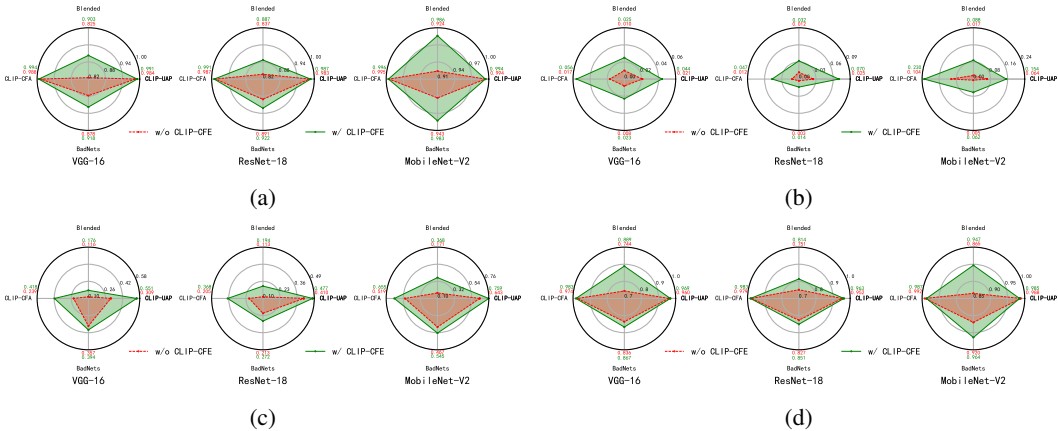

Figure 3: Attack success rate (ASR) of the **(a):** number-constrained backdoor attacks, **(b):** clean-label single-class attack (the access category $Y'$ is set to $\{0\}$), **(c):** dirty-label single-class attack (the access category $Y'$ is set to $\{1\}$), and **(d):** domain-constrained backdoor attacks (domain rate is set to 0) on the CIFAR-100 dataset. The red points represents w/o CLIP-based Clean Feature Erasing (CLIP-CFE), while the green points represents w/ CLIP-CFE. All experiments are repeated 5 times, and the results are computed withthe mean of five different runs.

### 4.2.1 CLIP-BASED CONTRASTIVE FEATURE AUGMENTATION

While the CLIP-UAP method has shown impressive results in terms of poisoning efficiency, it requires customization for different attack-target labels. In this section, we propose a more versatile trigger design that is independent of the attack-target label, enhancing the poisoning feature. Drawing inspiration from the entanglement between benign and poisoning features discussed in Sec. 3.2, we utilize contrastive optimization to augment the poisoning feature. Our expectation is that the poisoning feature extracted from the designed trigger will be more expressive compared to the clean feature extracted from the clean samples. Specifically, given a trigger $\delta_{con}^{t+1}$ to be optimized, two

random views (query: $x + \delta_{\text{con}}$ and key: $x_1 + \delta_{\text{con}}$) are created by different clean samples ($x$ and $x_1$). Positive pair is defined as such query-key pair, between different poisoning samples. Negative pairs are defined as pairs between poisoning example and its corresponding clean example, *i.e.* between $x + \delta_{\text{con}}$ and $x$. All views are passed through the pre-trained image encoder $\hat{\mathcal{E}}_i(\cdot)$ of the CLIP to acquire the representation $v$:

$$v_q = \hat{\mathcal{E}}_i(x + \delta_{\text{con}}), \quad v_+ = \hat{\mathcal{E}}_i(x_1 + \delta_{\text{con}}), \quad v_- = \hat{\mathcal{E}}_i(x). \tag{8}$$

CLIP-Contrastive feature augmentation (CLIP-CFA) focuses on optimizing the general trigger by maximizing the similarity between positive pairs while ensuring dissimilarity between negative pairs. To achieve this, we design a loss function as follows:

$$L_{\text{con}}(x, x_1, \delta_{\text{con}}) = -\frac{\langle v_q, v_+ \rangle}{\langle v_q, v_- \rangle}, \tag{9}$$

where $\langle \cdot, \cdot \rangle$ represents the cosine similarity between two vectors and the $\delta_{\text{con}}$ is optimized with:

$$\delta_{\text{con}} = \underset{||\delta_{\text{con}}||_p \leq \epsilon}{\arg\min} \sum_{(x,y) \in \mathcal{D}'} L_{\text{con}}(x, x_1, \delta_{\text{con}}). \tag{10}$$

Similar to Eq. 14, we also adopt the first-order optimization method PGD Madry et al. (2017) to solve the constrained minimization problem as follows:

$$\delta_{\text{con}}^{t+1} = \prod_\epsilon \left( \delta_{\text{con}}^t - \alpha \cdot \text{sign}(\nabla_{\delta_{\text{con}}} L_{\text{con}}(x, x_1, \delta_{\text{con}})) \right). \tag{11}$$

Therefore, the optimization should also be accumulated on all samples in the accessible clean training set $\mathcal{D}'$. Finally, the CLIP-CFA of set $\mathcal{D}'$ can be formulated as $\delta_{\text{con}} = \delta_{\text{con}}^T$, and the poison generator is formulated as $\mathcal{T}(x, \delta_{\text{con}}) = x + \delta_{\text{con}}$.

## 4.3 ATTACK SUMMARY

we present two independent trigger design methods: CLIP-UAP (Sec. A.6) and CLIP-CFA (Sec. 4.2.1). These triggers are aimed at enhancing the expression of poisoning features and can replace previous trigger design approaches, leading to improved performance in data-constrained backdoor attacks. Additionally, in Sec. 4.1, we introduce a CLIP-CFE method. This approach minimizes the influence of clean features during the poisoning process and can be integrated into any aforementioned trigger design methods. By combining trigger design and clean feature erasing, our final approach achieves state-of-the-art performance in all types of data-constrained backdoor attacks.

## 5 EXPERIMENTS

We provides an overview of the experimental settings, covering datasets, model architecture, evaluation metrics, baselines, and implementations (Appendix A.7). Subsequently, we perform comprehensive experiments to assess the effectiveness of our proposed methods through answering the following research questions: **RQ1: Are proposed technologies effective on three backdoor attacks?** (Sec. 5.1). **RQ2: Are proposed technologies harmless for Benign Accuracy?** (Sec. 5.2). **RQ3: Are proposed technologies stealthy for victims?** (Sec. A.8). **RQ4: Are proposed technologies effective for different poisoning settings?** (Sec. 5.3). In this section, we present the results specifically for CIFAR-100 datasets. Experimental outcomes for CIFAR-10 and ImageNet-50 are provided in Appendix A.9 and Appendix A.10 respectively. Additionally, for more experiments and further discussions, please refer to Appendix A.11, A.12, A.13, A.14, and A.15.

### 5.1 RQ1: ARE PROPOSED TECHNOLOGIES EFFECTIVE ON THREE BACKDOOR ATTACKS?

To assess the effectiveness of our proposed technologies, we conduct attacks on various target models and datasets, evaluating the Attack Success Rate (ASR) for each target model. In order to establish a basis for comparison, we introduce two baseline attack methods: BadNets Gu et al. (2019) and Blended Chen et al. (2017), as discussed in Sec. A.2.1. Fig. 3 illustrates the performance of the following types of backdoor attacks on the CIFAR-100 dataset: (a) number-constrained, (b)

Table 1: The Benign Accuracy (BA) on the CIFAR-100 dataset. All results are computed the mean by 5 different run.

| Trigger | Clean Feature Suppression | Backdoor Attacks | | | | | | | | | | | | Average |
|---|---|---|---|---|---|---|---|---|---|---|---|---|---|---|
| | | Number Constrained | | | Class Constrained ($Y' = \{0\}$) | | | Class Constrained ($Y' = \{1\}$) | | | Domain Constrained | | | |
| | | V-16 | R-18 | M-2 | V-16 | R-18 | M-2 | V-16 | R-18 | M-2 | V-16 | R-18 | M-2 | |
| BadNets | w/o CLIP-CFE | 0.698 | 0.728 | 0.722 | 0.698 | 0.730 | 0.728 | 0.700 | 0.728 | 0.729 | 0.699 | 0.727 | 0.728 | 0.718 |
| | w/ CLIP-CFE | 0.700 | 0.730 | 0.728 | 0.701 | 0.731 | 0.723 | 0.698 | 0.730 | 0.726 | 0.701 | 0.730 | 0.724 | 0.719 |
| Blended | w/o CLIP-CFE | 0.700 | 0.727 | 0.722 | 0.700 | 0.726 | 0.725 | 0.701 | 0.729 | 0.723 | 0.698 | 0.729 | 0.725 | 0.717 |
| | w/ CLIP-CFE | 0.700 | 0.730 | 0.727 | 0.701 | 0.729 | 0.727 | 0.699 | 0.730 | 0.724 | 0.700 | 0.731 | 0.727 | 0.719 |
| CLIP-UAP | w/o CLIP-CFE | 0.702 | 0.730 | 0.727 | 0.702 | 0.729 | 0.727 | 0.701 | 0.730 | 0.725 | 0.702 | 0.731 | 0.729 | 0.720 |
| | w/ CLIP-CFE | 0.700 | 0.731 | 0.725 | 0.702 | 0.732 | 0.726 | 0.699 | 0.732 | 0.724 | 0.700 | 0.730 | 0.725 | 0.719 |
| CLIP-CFA | w/o CLIP-CFE | 0.703 | 0.731 | 0.727 | 0.701 | 0.730 | 0.725 | 0.701 | 0.730 | 0.727 | 0.700 | 0.731 | 0.727 | 0.719 |
| | w/ CLIP-CFE | 0.702 | 0.729 | 0.729 | 0.701 | 0.730 | 0.727 | 0.702 | 0.731 | 0.725 | 0.702 | 0.730 | 0.727 | 0.720 |

clean-label single-class (class-constrained), (c) dirty-label single-class (class-constrained), and (d) out-of-the-domain (domain-constrained)[1].

**CLIP-based poisoning feature augmentation is more effective than previous attack methods.** Our proposed methods, CLIP-UAP and CLIP-CFA, outperform the baseline techniques (BadNets Gu et al. (2019) and Blended Chen et al. (2017) [2]) in terms of consistency across different attacks and target models. Specifically, we achieved an ASR of 0.878, 0.825, 0.984, and 0.988 for BadNets, Blended, CLIP-UAP, and CLIP-CFA, respectively, in the number-constrained backdoor attack on the VGG-16 model. These results provide evidence that our proposed poisoning feature augmentation generates more effective triggers compared to other methods.

**CLIP-based Clean Feature Suppression is useful for different attack methods.** Our proposed method, CLIP-CFE, has shown significant improvements in effectiveness compared to the baseline method without CLIP-CFE. In various cases, CLIP-CFE has enhanced the poisoning efficiency significantly. For instance, in the clean-label single-class backdoor attack on the VGG-16 dataset, we observed remarkable improvements of 187%, 150%, 110%, and 229% for BadNets, Blended, CLIP-UAP, and CLIP-CFA, respectively. However, it is worth noting that in the results of the domain-constrained backdoor attacks on MobileNet-V2 (as depicted in the right part of Fig. 3 (d)), CLIP-CFA and CLIP-UAP only slightly outperform the corresponding methods with CFE.

**More discussion.** While our technologies have shown significant improvements in poisoning efficiency compared to baselines, there are still important discussions that need to be addressed. We aim to provide answers to the following questions in a systematic manner in Appendix A.15: i) Why do we observe performance degradation in the clean-label single-class attack? ii) Why are domain-constrained backdoor attacks generally easier compared to class-constrained backdoor attacks?

## 5.2 RQ2: ARE PROPOSED TECHNOLOGIES HARMLESS FOR BENIGN ACCURACY?

As shown in Table 1, our CLIP-UAP and CLIP-CFA exhibit similar or even better average Benign Accuracy (BA) compared to the baseline methods, BadNets Gu et al. (2019) and Blended Chen et al. (2017). Additionally, it is worth noting that our proposed method, CLIP-CFE, does not negatively impact BA. This finding confirms that our technologies are harmless to the benign accuracy compared to baseline methods, even under various settings and different backdoor attacks.

## 5.3 RQ4: ARE PROPOSED TECHNOLOGIES EFFECTIVE FOR DIFFERENT POISONING SETTINGS?

**Experiments on different poison rates for number-constrained backdoor attacks.** We conduct ablation studies to assess the effectiveness of our proposed methods in reducing the number of

---

[1]Both the clean-label single-class and dirty-label single-class backdoor attacks represent extreme scenarios of the class-constrained backdoor attack. In the clean-label single-class attack, the targeted class category is set to $Y' = k$, while in the dirty-label single-class attack, it is set to $Y' = c$ where $c \neq k$. Similarly, the out-of-the-domain backdoor attack is an extreme scenario of the domain-constrained backdoor attack, with a domain rate of 0. For further details, please refer to Appendix A.7.

[2]While there are several more effective techniques for poisoning attacks, they typically necessitate access to the entire training data, rendering them unsuitable for our data-limited backdoor attacks.

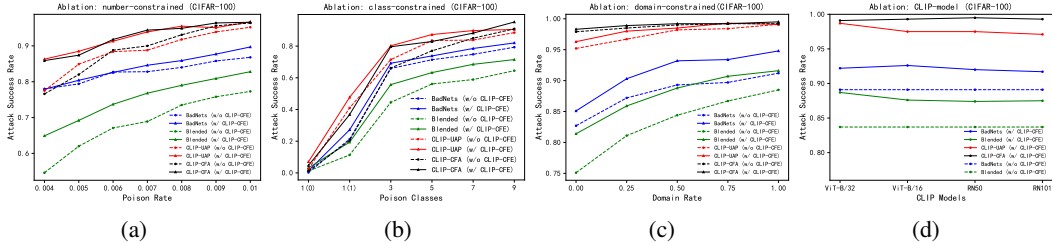

Figure 4: The ablation studies on the CIFAR-100 dataset. All results were computed as the mean of five different runs. **(a):** The ASR with different poisoning rates. **(B):** The ASR with different accessible class of poisoning samples, where 1 (0) and 1 (1) in the abscissa represent the clean-label and dirty-label single-class attacks, respectively. **(C):** The ASR with different domain rates. **(D):** The ASR across different pre-trained CLIP models for number-constrained backdoor attacks.

poisoning samples (poisoning rates) for number-constrained backdoor attacks. The results depicted in Fig. 4 (a) demonstrate the following: i) The attack success rate increases with higher poisoning rates for different attacks. ii) Our proposed CLIP-UAP and CLIP-CFA outperform the baseline techniques, BadNets Gu et al. (2019) and Blended Chen et al. (2017). iii) The incorporation of our proposed CLIP-CFE further enhances the poisoning effectiveness across different triggers.

**Experiments on different poison classes for class-constrained backdoor attacks.** We conduct ablation studies to assess the effectiveness of our proposed methods in increasing the number of poisoning classes for class-constrained backdoor attacks. The results presented in Fig. 4 (b) demonstrate the following: i) The attack success rate increases with higher poisoning classes for different attacks. ii) The attack success rate of clean-label single-class attack is lower than that of dirty-label single-class attacks. iii) Our proposed methods, CLIP-UAP and CLIP-CFA, outperform the baseline techniques, BadNets Gu et al. (2019) and Blended Chen et al. (2017). iv) The incorporation of our proposed CLIP-CFE further enhances the poisoning effectiveness across different triggers.

**Experiments on different domain rates for domain-constrained backdoor attacks.** We conduct ablation studies to assess the effectiveness of our methods in increasing the domain rate for domain-constrained backdoor attacks. The results depicted in Fig. 4 (c) demonstrate the following: i) The ASR increases with higher domain rates for different attacks. ii) Our proposed CLIP-UAP and CLIP-CFA outperform the baseline techniques, BadNets Gu et al. (2019) and Blended Chen et al. (2017). iii) The incorporation of our proposed CLIP-CFE further enhances the poisoning effectiveness across different triggers.

**Experiments on different large pre-trained models.** We utilize the pre-trained CLIP model as the basis for our technologies. It's worth noting that the community has proposed various CLIP variants. Therefore, an important practical consideration is whether our proposed technologies remain robust when applied different pre-trained CLIP models. To investigate this, we conduct ablation studies on different CLIP models for number-constrained backdoor attacks, as depicted in Fig. 4 (d). The results demonstrate that our proposed technologies exhibit robustness across different CLIP models, with ViT-B/32 emerging as a competitive choice for all methods.

## 6 CONCLUSION

In this paper, we address the challenges of data-constrained backdoor attacks, which occur in more realistic scenarios where victims collect data from multiple sources and attackers cannot access the full training data. To overcome the performance degradation observed in previous methods under data-constrained backdoor attacks, we propose three technologies from two streams that leverage the pre-trained CLIP model to enhance the efficiency of poisoning. Our goal is to inspire the research community to explore these realistic backdoor attack scenarios and raise awareness about the threats posed by such attacks. In the Sec A.16, we discuss the limitations of our approach and outline potential future directions for backdoor learning research.

## ACKNOWLEDGMENTS

This work was funded by the National Natural Science Foundation of China (U19B2044). Sponsored by Zhejiang Lab Open Research Project (NO. K2022QA0AB04). Supported by the Fundamental Research Funds for the Central Universities.

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

# A  APPENDIX

## CONTENTS

## A.1 DETAILED BACKGROUND AND RELATED WORK

### A.1.1 POISONING EFFICIENCY IN BACKDOOR ATTACKS

Existing studies aim at improving the poisoning efficiency of backdoor attacks can be categorized into two main areas.

**Designing Efficient Triggers** The design of efficient triggers that are easier for DNNs to learn has garnered significant interest. Researchers have recently drawn inspiration from Universal Adversarial Perturbations (UAPs) Moosavi-Dezfooli et al. (2017) and optimized UAPs on pre-trained clean models to create effective triggers, which have been widely utilized in various studies Zhong et al. (2020); Li et al. (2020); Doan et al. (2021). However, this approach requires a pre-trained clean model on the training set, which is not practical for data-constrained backdoor attacks.

**Selecting Efficient Poisoning Samples** Efficient sample selection for poisoning attacks is a critical yet under-explored aspect that is distinct from trigger design. Xia et al. (2022b) were among the first to investigate the contribution of different data to backdoor injection. Their research revealed that not all poisoning samples contribute equally, and appropriate sample selection can greatly enhance the efficiency of data in backdoor attacks. Additionally, various studies Li et al. (2023a;d); Gao et al. (2023); Guo et al. (2023) follow this setting, and the sample efficiency has been further improved.

### A.1.2 POISONING STEALTHINESS IN BACKDOOR ATTACKS

Existing studies focused on increasing the stealthiness of backdoor attacks can be categorized into two main areas.

**Designing Invisible Triggers** The concept of invisible triggers aims to ensure that poisoning images are visually indistinguishable from clean samples, thus evading detection in both pixel and feature spaces. This perspective is the most straightforward approach to bypass defenses. Chen et al. (2017) first propose a blended strategy to evade human detection by blending clean samples with the trigger to create poisoning samples. Subsequent studies Zhong et al. (2020); Li et al. (2020); Doan et al. (2021) focus on constraining the norm of the trigger through optimization methods. Moreover, some studies have explored the use of natural patterns such as warping Nguyen & Tran (2021), rotation Wu et al. (2022), style transfer Cheng et al. (2021b), frequency Feng et al. (2022); Zeng et al. (2021), and reflection Liu et al. (2020) to create triggers that are more imperceptible to human inspection. In contrast to previous works that employ universal triggers, Li et al. (2021c) employ GAN models to generate sample-specific triggers, which are similar to adversarial examples and extremely imperceptible to humans.

**Clean-label Attacks.** Clean-label attacks refer to backdoor attacks where the target labels of the poisoning samples align with their perception labels. Turner et al. (2019) is the first to explore clean-label attacks by employing GAN-based and adversarial-based perturbations. Compared to standard backdoor attacks, clean-label attacks are typically less effective due to the model's tendency to associate natural features, rather than backdoor triggers, with the target class. Recent studies have focused on aligning features Saha et al. (2020) or gradients Souri et al. (2021) between perturbed inputs from the target class and trigger-inserted inputs from the non-target class through pretraining on the entire training set. Additionally, some study Zeng et al. (2022) has proposed optimizing the backdoor trigger using only the knowledge about the target-class training data. In this approach, the trigger is optimized to point towards the interior of the target class, resulting in improved effectiveness.

### A.1.3 CONTRASTIVE LANGUAGE-IMAGE PRE-TRAINING (CLIP) MODEL

Our method introduces the Contrastive Language-Image Pre-Training (CLIP) Radford et al. (2021) model into backdoor injection and we introduce it here. CLIP is a revolutionary deep learning model developed by OpenAI that is designed to connects texts and images by bringing them closer in a shared latent space, under a contrastive learning manner. The CLIP model is pre-trained on 400 million image-text pairs harvested from the Web, containing two encoder: CLIP text encoder $\hat{\mathcal{E}}_t(\cdot)$ and CLIP image encoder $\hat{\mathcal{E}}_i(\cdot)$. These encoders project the text and image to the CLIP common embedded feature space. Since natural language is able to express a much wider set of visual concepts, it contains ability to generalize across a wide range of tasks and domains, such as text-driven image manipulation Patashnik et al. (2021), zero-shot classification Cheng et al. (2021a), domain generalization Niu et al. (2022). To our best knowledge, our paper is the first study to explore the usage of CLIP model in the security community.

### A.2 DETAILED PIPELINE OF DATA-CONSTRAINED BACKDOOR ATTACKS.

### A.2.1 EXAMPLES OF BACKDOOR ATTACKS IN OUR STUDY

Here, we present three popular backdoor attack methods that serve as the baseline for our preliminary experiments, providing insight into the motivation discussed in Sec. 3. All attacks follow the pipeline described in Sec. 2.2.

**BadNets.** BadNets Gu et al. (2019) is the pioneering backdoor attack in deep learning and is often used as a benchmark for subsequent research. It utilizes a $2 \times 2$ attacker-specified pixel patch as the universal trigger pattern attached to benign samples.

**Blended.** Chen et al. (2017) first discuss the requirement for invisibility in backdoor attacks. They propose that the poisoning image should be visually indistinguishable from its benign counterpart to evade human inspection. To meet this requirement, they introduce a blending strategy where poisoning images are created by blending the backdoor trigger with benign images. Formally, the poison generator can be formulated as $\mathcal{T}(x, t) = \lambda \cdot t + (1 - \lambda) \cdot x$, where $\lambda$ represents the blend ratio (we set $\lambda = 0.15$ for all experiments in this paper), and $t$ is an attacker-specified benign image serving as the universal trigger pattern.

**Universal Adversarial Perturbations (UAP).** Inspired by Universal Adversarial Perturbations (UAPs) in adversarial examples, some studies Zhong et al. (2020); Li et al. (2020); Doan et al. (2021) propose optimizing a UAP on a pre-trained clean model as the natural trigger, formulated as $\mathcal{T}(x, t) = x + t$, where $t$ is a pre-defined UAP serving as the universal trigger pattern. It's worth noting that UAP-based backdoor attacks require a clean model pre-trained on the entire training set, which is not suitable for the discussed settings. However, to better explain our motivation that previous technologies exhibit significant performance degradation in data-constrained backdoor attacks, we assume the availability of a clean model pre-trained on the original training dataset in this section. It is important to acknowledge that this assumption does not hold in an actual attack scenario.

### A.2.2 EXPERIMENTAL SETTINGS.

To evaluate the performance of three backdoor attack methods (BadNets, Blended, and UAP) under data-constrained scenarios, we conduct experiments on the CIFAR-10 dataset. Specifically, we consider three types of data constraints: number, class, and domain. The settings for the poisoning attacks follow those described in Sec. A.2.1. In all attacks, we set the attack-target label $k$ to category 0. For our experiments, we select the VGG-16 model as the victim model and employ SGD as the optimizer with a weight decay of 5e-4 and a momentum of 0.9. The batch size is set to 256, and the initial learning rate is set to 0.01. The learning rate is multiplied by 0.1 at the 35-th and 55-th epochs, and the training is conducted for a total of 70 epochs.

### A.2.3 NUMBER-CONSTRAINED BACKDOOR ATTACKS

**Definition.** Let $\mathcal{D}'$ denote the data manipulable by the malicious source, and $\mathcal{D}$ represent all the data available to the data collector. In the number-constrained scenario, as illustrated in Fig. 5 (a),

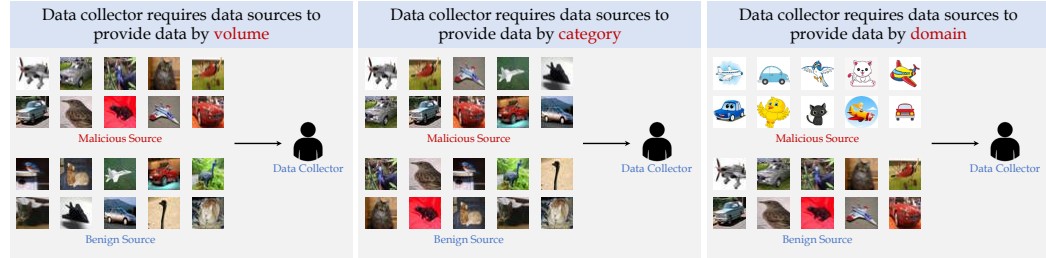

| (a) Number-constrained Scenario | (b) Class-constrained Scenario | (c) Domain-constrained Scenario |

Figure 5: Three data-constrained attack scenarios, where the data provided by each data source is independently and identically distribution in number-constrained backdoor attacks, each data source provides data belonging to different categories in class-constrained backdoor attacks, and each data source provides data from different domains in domain-constrained backdoor attacks.

the data collector gathers data from multiple sources, including both malicious and benign sources, to form $\mathcal{D}$. The data provided by each data source is independently and identically distributed. In other words, $\mathcal{D}$ and $\mathcal{D}'$ belong to the same distribution, but in terms of quantity, $N' < N$. The setting of number-constrained backdoor attacks is similar to that of data-efficient backdoor attacks discussed in previous studies Xia et al. (2022b); Zhong et al. (2020). Both aim to improve the Attack Success Rate (ASR) under a low poisoning rate. However, previous studies assumed that the attacker has access to the entire training set $\mathcal{D}$, which enables efficient trigger design and sample selection. For example, some studies Zhong et al. (2020) draw inspiration from Universal Adversarial Perturbations (UAPs) in adversarial examples and propose to optimize a UAP on a clean model pre-trained on the training set as the natural trigger. Xia *et al.* Xia et al. (2022b) enhance the poisoning efficiency in backdoor attacks by selecting poisoning data from the entire training set. Although these methods have achieved remarkable results, they cannot be directly applied to number-constrained backdoor attacks due to the lack of access to the entire training set.

**Experimental results.** In this section, we investigate the performance degradation of previous studies in number-constrained backdoor attacks. As shown in Fig. 2 (a), the attack success rate experiences a significant decrease as the number ($P$) of poisoning samples decreases, particularly for Blended backdoor attacks. It is worth noting that Universal Adversarial Perturbations (UAP) achieves relatively favorable results even with a low poisoning rate. This can be attributed to the utilization of a proxy model that is pre-trained on the entire training set ($\mathcal{D}$). However, in our settings, UAP is not accessible, and we present the results for UAP to effectively demonstrate the performance degradation even when a pre-trained proxy model is available.

### A.2.4 CLASS-CONSTRAINED BACKDOOR ATTACKS

**Definition.** In the class-constrained scenario, let $\mathcal{D}'$ represent the data manipulable by the malicious source, and $\mathcal{D}$ denote all the data available to the data collector. As depicted in part (b) of Fig. 5, the data collector gathers data from multiple sources, including both malicious and benign sources, to form $\mathcal{D}$. Each data source provides data belonging to different categories, resulting in $\mathcal{D}'$ containing only a subset of categories present in $\mathcal{D}$. Therefore, $\mathcal{D}$ and $\mathcal{D}'$ follow distinct distributions. More specifically, the accessible clean training set $\mathcal{D}' \subset X \times Y'(\mathcal{D}' = \{(x_i, y_i)|i = 1, \cdots, N'\})$ is a subset of the entire training set $\mathcal{D} \subset X \times Y(\mathcal{D} = \{(x_i, y_i)|i = 1, \cdots, N\})$, where $Y' \subset Y = \{1, 2, \cdots, C\}$. Class-constrained backdoor attacks can be seen as a general setting of clean-label backdoor attacks Turner et al. (2019); Saha et al. (2020); Souri et al. (2021). In clean-label backdoor attacks, the accessible clean training set $\mathcal{D}'$ is defined as $\mathcal{D}' \subset X \times Y'$, where $Y' = \{k\}$ and $k$ represents the attack-target label.

**Experimental results.** In this section, we explore the performance degeneration of previous studies in class-constrained backdoor attacks. As illustrated in Fig. 2 (b), attack success rate decreases as the number of class ($C'$) in the poisoning set decreases, which is the similar as experimental results on the number-constrained backdoor attacks.

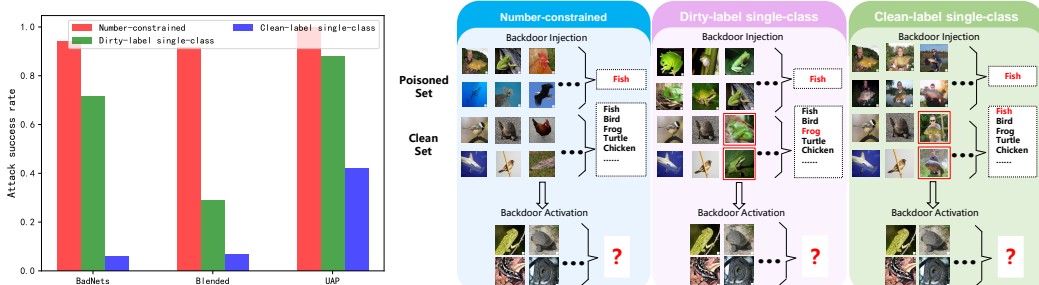

(a) ASR of three backdoor attacks ($p = 1000$ for all experiments).

(b) Visualizations of the backdoor injection and activation phases for three backdoor attacks.

Figure 6: Analyses the entanglement between benign and poisoning features on the number-constrained, dirty-label single-class, and clean-label single-class backdoor attacks.

### A.2.5 DOMAIN-CONSTRAINED BACKDOOR ATTACKS

**Definition.** In the domain-constrained scenario, as depicted in part (c) of Fig. 5, the data collector gathers data from multiple sources (both malicious and benign) to form $\mathcal{D}$. Each data source provides data from a different domain, resulting in $\mathcal{D}'$ containing only a subset of the domains present in $\mathcal{D}$. Consequently, $\mathcal{D}$ and $\mathcal{D}'$ belong to different distributions. We examine an extreme scenario in domain-constrained backdoor attacks, where the test dataset follows the same distribution as the benign source ($\mathcal{D} \setminus \mathcal{D}'$) and is outside the domain of the malicious source $\mathcal{D}' \subset X \times Y'$ ($\mathcal{D}' = \{(x_i, y_i)|i = 1, \cdots, N'\}$). Evidently, images belonging to the same class can stem from diverse distributions, which we term as domain distinctions. To illustrate, consider the "Car" category found in both the ImageNet and CIFAR-10 datasets. Despite sharing this category, they exhibit disparate distributions, classifying them into distinct domains. In our context, characterized as "domain-constrained" backdoor attacks, the domain pertains to the differing distributions within the same image class. This delineation underpins a tangible attack scenario. For instance, consider a victim endeavoring to train a comprehensive classifier capable of generalizing across varied data distributions. However, the assailant lacks comprehensive knowledge of the domains from which the victim sources data; thus, their capacity is confined to contaminating images within one or several domains, among the array of domains within the training set.

**Experimental results.** To simulate the domain-constrained scenario, we conducted experiments with the following settings in this section: we designate the CIFAR-10 dataset as the benign source, the ImageNet dataset as the malicious source, and evaluated the attack performance on the CIFAR-10 dataset. Fig. 2 (c) illustrates the results, showing a decrease in the attack success rate as the domain rate (the proportion of poisoning sets sampled from $\mathcal{D} \setminus \mathcal{D}'$ and $\mathcal{D}'$) in the poisoning set decreases. This observation aligns with the experimental findings in the number-constrained and class-constrained backdoor attacks.

### A.3 ANALYZING THE ENTANGLEMENT BETWEEN BENIGN AND POISONING FEATURES

In this section, we provide three observations on data-constrained backdoor attacks to demonstrate the entanglement between benign and poisoning features does exist in the backdoor injection process, and it is the main reason why current attack methods fail in data-constrained scenarios.

**Observation 1: BadNet outperforms Blended notably in data-constrained attack scenarios.** In a practical experimentation setting, BadNet and Blended exhibit comparable performance under unrestricted attack conditions (the leftmost point on the horizontal axis of Fig. 2). Conversely, in data-constrained attack scenarios, BadNet outperforms Blended notably. This intriguing disparity requires elucidation. BadNet employs a $2 \times 2$ attacker-specified pixel patch as a universal trigger pattern attached to benign samples, whereas Blended employs an attacker-specified benign image for the same purpose. Comparatively, Blended's trigger exhibits greater feature similarity to benign images, engendering a more pronounced feature entanglement between poisoned and benign attributes. Accordingly, the performance of the blended dirty-label single-class scenario significantly lags be-

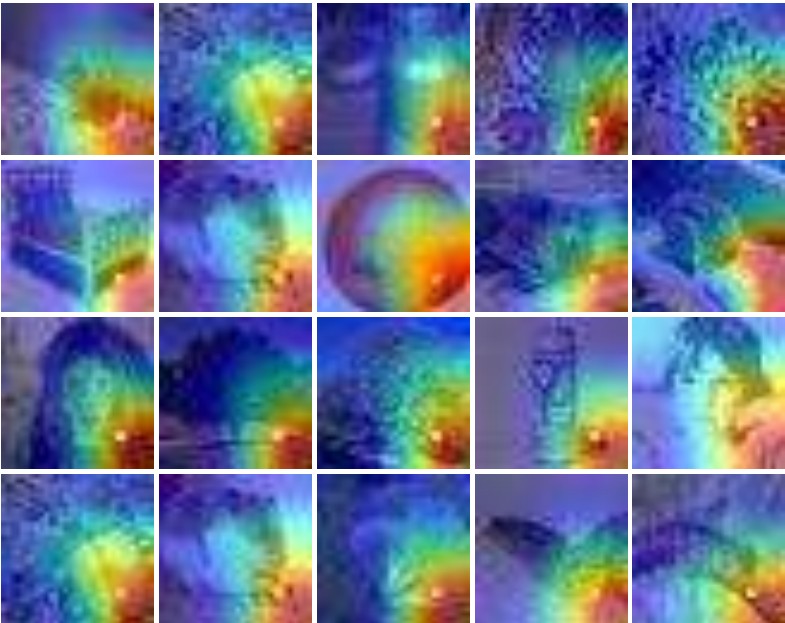

Figure 7: The Grad-CAM of the backdoor model on the CIFAR-100 dataset.

hind other cases, lending credence to our hypothesis that entanglement underpins the degradation of data-constrained backdoor attacks.

**Observation 2: Attack efficiency of number-constrained, dirty-label single-class, and clean-label single-class backdoor attacks decreases in turn under the same poison rate.** We further investigated our hypothesis and present our findings of entanglement in Fig. 6. As shown in Fig. 6 (a), the attack efficiency of number-constrained, dirty-label single-class, and clean-label single-class backdoor attacks[1] decreases in turn under the same poison rate. To understand the reason behind this phenomenon, we provide visualizations of the backdoor injection and activation phases for these three attacks in Fig. 6 (b). For the **number-constrained backdoor attack**, the distribution of poisoning samples (consisting of both benign and poisoning features) in the backdoor injection phase is the same as that in the backdoor activation phase. In other words, both benign and poisoning features are activated simultaneously during both phases. However, for the **dirty-label single-class backdoor attack**, the distribution of poisoning samples (consisting of single-class benign and poisoning features) in the backdoor injection phase is different from that in the backdoor activation phase. During the injection phase, both benign and poisoning features are activated, but during activation phase, only the poisoning feature is activated. This is the reason why previous attack methods on dirty-label single-class backdoor attacks exhibit performance degeneration. The **clean-label single-class backdoor attack** is similar to the dirty-label single-class backdoor attack in terms of the distribution of poisoning samples. However, during backdoor injection, there is competing activation[3] between benign and poisoning features. Consequently, the poisoning efficiency of clean-label single-class backdoor attacks is lower than that of dirty-label single-class backdoor attacks.

**Observation 3: Substantial dissimilarities in activation between poisoned samples that share the same trigger.** We have embraced Grad-CAM Selvaraju et al. (2017) to more effectively corrob-

---

[3]In the clean-label single-class backdoor attack, the benign feature of the accessible class (the same as the attack-target class) in both poisoning and clean sets is labeled with the same label (e.g., "Fish" in Fig. 6), and the clean set contains more samples of the attack-target class. As a result, the presence of the benign feature in the poisoning set hampers the activation of the poisoning features. In contrary, the benign feature of the accessible class in poisoning and clean sets is labeled with the different label in the dirty-label single-class backdoor attack (e.g., the benign feature in the clean set is labeled as "Frog", while the benign+poisoning feature in the poisoning set is labeled as "Fish"). Consequently, the benign feature in the poisoning set does not impact the activation of the poisoning features.

orate the presence of a correlation between the benign and the backdoor features. To substantiate this hypothesis, we have incorporated Grad-CAM outcomes to our study. Sgrad-campecifically, we have applied this technique to the BadNet-based poison model on the CIFAR-100 dataset, as depicted in Figure 7. The discernible results from these Grad-CAM visualizations underscore the substantial dissimilarities in activation between poisoned samples that share the same trigger. This visual evidence compellingly demonstrates the intricate entanglement existing between the benign and the backdoor features during instances of backdoor attacks.

## A.4 THREAT MODEL

**Attack scenario.** The proliferation of large-scale artificial intelligence models, such as ChatGPT and Stable Diffusion, necessitates the collection of massive amounts of data from the web. However, the security and trustworthiness of this data cannot always be guaranteed. This data collection pipeline inadvertently introduces vulnerabilities that can be exploited by data-based backdoor attacks. Attackers can strategically inject poisoning data into the training dataset and publish it on the internet, potentially compromising the integrity and performance of these models. Unlike previous attack scenarios where all training data is sourced from a single provider, we consider a more realistic scenario in which victims collect data from multiple sources. In this scenario, attackers only have access to a portion of the training dataset. This situation mirrors the real-world training process of models that utilize diverse public data. By acknowledging the challenges posed by multi-source data collection and limited attacker access, our study provides valuable insights into the security implications of such scenarios.

**Attack goal.** The objective of our paper is aligned with popular backdoor attacks, as seen in previous studies Gu et al. (2019); Li et al. (2021a). The attackers aim to activate a hidden trigger within the model by providing specific inputs, leading the model to produce incorrect results. Our attack strategy emphasizes three key properties: (i) *Minimal side effects:* The backdoor attacks should not adversely impact the accuracy of the model on benign inputs. (ii) *Effective backdoor:* The attack should have a high success rate across various datasets and models, ensuring its efficiency. (iii) *Stealthy attack:* The backdoor attacks should be inconspicuous and difficult to detect, maintaining their stealthiness. Our research aims to develop invigorative backdoor attacks that strike a balance between effectiveness and preserving the integrity of the model's performance on legitimate inputs.

**Attackers' prior knowledge.** In order to simulate a realistic scenario, we assume that the attackers have no access to the models or training details. They possess only general knowledge about the class labels involved in the task. This assumption reflects a more challenging and practical setting, where attackers have limited information about the target system.

**Attackers' capabilities.** Building upon previous studies Gu et al. (2019), we make the assumption that the attackers possess the capability to control the training data. However, we further impose a stricter assumption in this work, stating that the attackers have control over only a portion of the training data. Consequently, we divide the attack scenario into three distinct tasks, each representing different capabilities of the attacker. These tasks include: (i) *Number-constrained backdoor attacks*, where the attacker has access to only a subset of the training data; (ii) *Class-constrained backdoor attacks*, where the attacker has access to only a subset of the classes in the training data; and (iii) *Domain-constrained backdoor attacks*, where the attacker has access to only a subset of the domains within the training data. By considering these various constraints, we provide a comprehensive analysis of backdoor attacks in different data-constrained scenarios.

## A.5 CLIP FOR ZERO-SHOT CLASSIFICATION

The pre-trained CLIP model Radford et al. (2021) possesses the ability to express a broader range of visual concepts and has been utilized as a general feature extractor in various tasks. These tasks include text-driven image manipulation Patashnik et al. (2021), zero-shot classification Cheng et al. (2021a), and domain generalization Niu et al. (2022). In this section, we introduce the pipeline of CLIP in zero-shot classification, which can serve as inspiration for incorporating it into our clean feature erasing approach. CLIP achieves zero-shot classification by aligning text and image features. Firstly, CLIP employs its text encoder, denoted as $\hat{\mathcal{E}}_t(\cdot)$, to embed the input prompts ("a photo of a $c_i$") into text features $T_i \in \mathbb{R}^d$, where $i = \{1, 2, \cdots, C\}$ represents the classes. Subsequently, the image feature $I_j \in \mathbb{R}^d$ of image $x_j$ is embedded using the image encoder, denoted as $\hat{\mathcal{E}}_i(\cdot)$. During

the inference phase, the classification prediction $y_j$ is computed using the cosine similarity between $T_i$ and $I_j$. This can be expressed as:

$$y_j = \arg\max_i(\langle I_j, T_i \rangle), \quad i \in \{1, 2, \cdots, C\}, \tag{12}$$

where $C$ represents the number of classes, and $\langle \cdot, \cdot \rangle$ represents the cosine similarity between two vectors.

### A.6 CLIP-BASED UNIVERSAL ADVERSARIAL PERTURBATIONS

In this section, we also employ the widely-used pre-trained CLIP model Radford et al. (2021) to generate universal adversarial perturbations as the backdoor trigger. Xia et al. (2023) argue that deep models inherently possess flaws, and it is easier to exploit and enhance an existing flaw to serve as a backdoor rather than implanting a new one from scratch (BadNets Gu et al. (2019) and Blended Chen et al. (2017)). Universal Adversarial Perturbations (UAP) Zhong et al. (2020); Xia et al. (2023) utilize these inherent flaws in models as triggers, providing a straightforward method for augmenting the poisoning feature. However, this approach typically requires a feature extractor that has been pre-trained on the entire training set, which is not practical in data-constrained backdoor attacks. To address this limitation, we propose a CLIP-based Universal Adversarial Perturbations (CLIP-UAP) method. Specifically, given an accessible clean training set $\mathcal{D}' = \{(x_i, y_i)|i = 1, \cdots, N'\}$ and an attack-target label $k$, the defined trigger can be formulated as follows:

$$\delta_{\text{uap}} = \arg\min_{||\delta_{\text{uap}}||_p \leq \epsilon} \sum_{(x,y) \in \mathcal{D}'} L(f_{CLIP}(x + \delta_{\text{uap}}, \mathbb{P}), k), \tag{13}$$

where $\mathbb{P}$ and $f_{CLIP}$ are defined as shown in Eq. 4 and Eq. 5, respectively. Similar to Eq. 6, we utilize the first-order optimization method known as Projected Gradient Descent (PGD) Madry et al. (2017) to solve the constrained minimization problem. The optimization process can be expressed as follows:

$$\delta_{\text{uap}}^{t+1} = \prod_{\epsilon} \left( \delta_{\text{uap}}^t - \alpha \cdot \text{sign}(\nabla_{\delta_{\text{uap}}} L(f_{CLIP}(x + \delta_{\text{uap}}^t, \mathbb{P}), k)) \right), \tag{14}$$

where $t$, $\nabla_{\delta_{\text{uap}}} L(f_{CLIP}(x + \delta_{\text{uap}}^t, \mathbb{P}), k)$, and $\prod$ hold the same meaning as in Eq. 6. Unlike the sample-wise clean feature erasing noise, the CLIP-UAP serves as a universal trigger for the entire training set. Therefore, it follows the optimization formulation presented in Eq. 14 to generate $\delta_{\text{uap}}^{t+1}$ at each step $t$. The optimization process is performed on all samples in the accessible clean training set $\mathcal{D}'$. Consequently, the CLIP-UAP for the set $\mathcal{D}'$ can be represented as $\delta_{\text{uap}} = \delta_{\text{uap}}^T$, and the poison generator is formulated as $\mathcal{T}(x, \delta_{\text{uap}}) = x + \delta_{\text{uap}}$.

### A.7 EXPERIMENTAL SETUP

#### A.7.1 DATASETS

We use the following three popular datasets in image classification:

**CIFAR-10 Krizhevsky et al. (2009).** CIFAR-10 is a tiny object classification dataset containing 50,000 training images and 10,000 testing images. Each image has a size of $32 \times 32 \times 3$ and belongs to one of 10 classes.

**CIFAR-100 Krizhevsky et al. (2009).** Similar to CIFAR-10, CIFAR-100 is also a tiny object classification dataset containing 50,000 training images and 10,000 testing images. Each image has a size of $32 \times 32 \times 3$ and belongs to one of 100 classes.

**ImageNet-50 Deng et al. (2009).** ImageNet is the most popular object classification dataset containing 1.3M training images and 50K testing images. Each image has a size of $224 \times 224 \times 3$ and belongs to one of 1000 classes. For simplicity, we randomly sampled 50 categories to compose a tiny dataset: ImageNet-50. Our ImageNet-50 dataset contains 60K training images and 2.5K testing images.

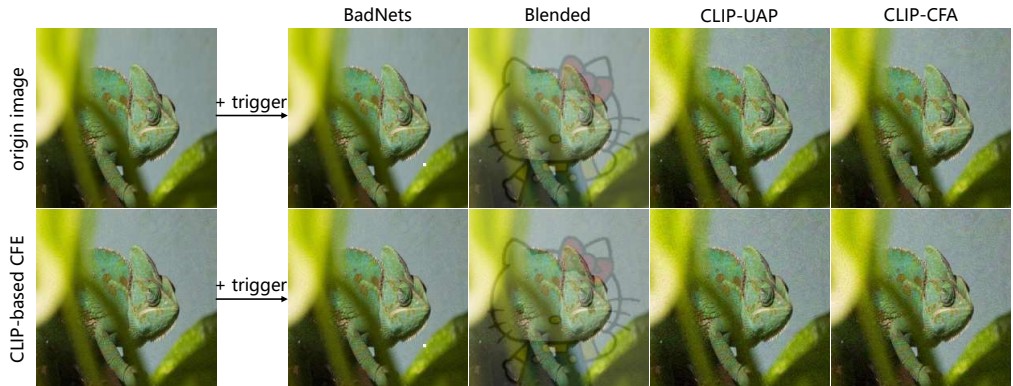

Figure 8: Visualizations of the poisoning samples with different triggers.

### A.7.2 Model Architecture.

We verify the performance on three popular model architectures of image classification: **VGG-16 Simonyan & Zisserman (2014)**, **ResNet-18 He et al. (2016)**, and **MobileNet-V2 Sandler et al. (2018)**. All of them are widely used in various areas of artificial intelligence, such as flower classification Xia et al. (2017), pulmonary image classification Wang et al. (2019b), fault diagnosis Wen et al. (2020), and Covid-19 screening Farooq & Hafeez (2020).

### A.7.3 Baseline and Comparison.

Our method contains two aspects: clean feature suppression and poisoning feature augmentation. Among them, poisoning feature augmentation can be accomplished through designing efficient and data-independent triggers, while clean feature suppression is orthogonal to previous trigger designing and can be integrated into any backdoor triggers. Therefore, we compare our two designed triggers CLIP-based universal adversarial perturbations (CLIP-UAP) and CLIP-based contrastive feature augmentation (CLIP-CFA) with two popular triggers: BadNets Gu et al. (2019)and Blended Chen et al. (2017). All of them are independent of the training data therefore can be implemented easily in the introduced data-constrained backdoor attacks. Although there are also contain other advanced clean-label backdoor attacks Liu et al. (2020); Barni et al. (2019); Saha et al. (2020); Souri et al. (2022) and state-of-the-art backdoor attacks Zhong et al. (2020), a substantial proportion of them Liu et al. (2020); Saha et al. (2020); Souri et al. (2022); Zhong et al. (2020) operate within the threat model that necessitates a proxy model pre-trained on the entire training set. This precondition becomes challenging to fulfill in the context of a many-to-one (M2O) data collection attack scenario. To verify the validity of the clean feature suppression, we integrate the proposed CLIP-based clean feature erasing (CLIP-CFE) onto currently designed triggers: two our designed triggers and two baseline triggers.

### A.7.4 Implementations.

In order to demonstrate the effectiveness of our proposed method, we conduct experiments on three datasets (CIFAR-10, CIFAR-100, and ImageNet-50). For the CIFAR-10 and CIFAR-100 datasets, we choose VGG-16, ResNet-18, and MobileNet-V2 as the victim models. All models use the SGD optimizer with a momentum of 0.9, weight decay of 5e-4, and a learning rate of 0.01 (0.1 for MobileNet-V2), which is multiplied by 0.1 at epoch 35 and 55. For the ImageNet-50 dataset, we use VGG-16 and MobileNet-V2 as the victim models. We use the SGD optimizer with a momentum of 0.9, weight decay of 5e-4, and a learning rate of 0.05 (0.01 for VGG-16), which is multiplied by 0.1 at epoch 35 and 55. The complete training epochs is 70.

In the number-constrained scenario, we conducted experiments with poisoning rates of 0.01 (P=500), 0.015 (P=750), and 0.007 (P=453) for the CIFAR10, CIFAR100, and ImageNet-50 threes datasets, respectively. In the class-constrained scenario, experiments with poisoning rates of 0.02 (P=1000), 0.01 (P=500), and 0.02 (P=1296) for three datasets. We choose two extreme scenario in the class-constrained backdoor attacks, denoted as clean-label single-class backdoor attack and dirty-label single-class backdoor attack. Specifically, the accessed class category is set to $Y' = \{k\}$

Table 2: The Peak Signal-to-noise Ratio (PSNR) and Structural Similarity Index (SSIM) on the ImageNet-50 dataset. All results are computed on the 500 examples.

| Metrics | Trigger | Clean Feature Suppression | Backdoor Attacks | | | |
|---|---|---|---|---|---|---|
| | | | Number Constrained | Clean-label single-class | Dirty-label single-class | Domain Constrained |
| PSNR (↑) | BadNets | w/o CLIP-CFE | 32.18 | 31.26 | 30.91 | 32.22 |
| | | w/ CLIP-CFE | 32.19 | 31.41 | 31.39 | 32.17 |
| | Blended | w/o CLIP-CFE | 21.68 | 21.37 | 20.90 | 21.60 |
| | | w/ CLIP-CFE | 21.67 | 21.31 | 20.99 | 21.58 |
| | CLIP-UAP | w/o CLIP-CFE | 33.11 | 33.66 | 32.84 | 32.64 |
| | | w/ CLIP-CFE | 33.10 | 33.64 | 32.81 | 32.59 |
| | CLIP-CFA | w/o CLIP-CFE | 32.74 | 32.54 | 32.70 | 32.46 |
| | | w/ CLIP-CFE | 32.72 | 32.52 | 32.67 | 32.41 |
| SSIM (↑) | BadNets | w/o CLIP-CFE | 0.995 | 0.995 | 0.995 | 0.995 |
| | | w/ CLIP-CFE | 0.995 | 0.995 | 0.995 | 0.995 |
| | Blended | w/o CLIP-CFE | 0.794 | 0.820 | 0.730 | 0.787 |
| | | w/ CLIP-CFE | 0.827 | 0.834 | 0.772 | 0.819 |
| | CLIP-UAP | w/o CLIP-CFE | 0.719 | 0.822 | 0.641 | 0.702 |
| | | w/ CLIP-CFE | 0.843 | 0.885 | 0.793 | 0.822 |
| | CLIP-CFA | w/o CLIP-CFE | 0.707 | 0.795 | 0.637 | 0.692 |
| | | w/ CLIP-CFE | 0.830 | 0.857 | 0.788 | 0.810 |

and $Y' = \{c\}, c \neq k$ for clean-label single-class backdoor attack and dirty-label single-class backdoor attack, respectively. In the domain-constrained scenario, experiments with poisoning rates of 0.02 (P=1000, 1000, and 1296) for three datasets. The out-of-domain samples of all experiments are selected from other ImageNet-1K datasets that are not ImageNet-50. The attack-target class $k$ is set to category 0 for all experiments of above three data-constrained backdoor attacks.

### A.7.5  EVALUATION METRICS.

We evaluate the performance of our method in terms of *Harmlessness:* **Benign Accuracy (BA)**, *Effectiveness:* **Attack Success Rate (ASR)**, and *Stealthiness:* **Peak Signal-to-noise Ratio (PSNR)** Huynh-Thu & Ghanbari (2008) and **Structural Similarity Index (SSIM)** Wang et al. (2004).

**Benign Accuracy (BA).** BA is the clean accuracy of the testing set $\mathcal{D}_t = \{(x_i, y_i)|i = 1, \cdots, M\}$ and is applied to evaluate the *Harmlessness* of the backdoor. When BA of the infected model is similar to the accuracy of the clean model, we believe that the current attack technique is harmless.

**Attack Success Rate (ASR).** ASR is applied to evaluate the *effectiveness* of the backdoor attack, which is the fraction of testing images with specific trigger that are predicted as the target class. Specifically, For $M'$ images in the testing set that do not belong to the attack-target class ($k$), the ASR is formulated as:

$$\text{ASR} = \frac{\sum_{i=1}^{M'} \mathbb{I}(f(\mathcal{T}(x_i, t); \Theta) = k)}{M'}, \quad (x_i, y_i) \in \mathcal{D}'_t, \tag{15}$$

where $\mathcal{D}'_t$ is a subset of testing set $\mathcal{D}_t$ ($\mathcal{D}'_t \subset \mathcal{D}_t$), containing the images whose label is not the attack-target class $k$.

**Peak Signal-to-noise Ratio (PSNR)** Huynh-Thu & Ghanbari (2008). PSNR is applied to measure the similarity between clean images and the corresponding poisoning images. Give a image $x_i \in \mathcal{D}_t$

Table 3: The Benign Accuracy (BA) on the CIFAR-10 dataset. All results are computed the mean by 5 different run.

| Trigger | Clean Feature Suppression | Backdoor Attacks | | | | | | | | | | | |
| | | Number Constrained | | | Class Constrained ($Y' = \{0\}$) | | | Class Constrained ($Y' = \{1\}$) | | | Domain Constrained | | |
| | | V-16 | R-18 | M-2 | V-16 | R-18 | M-2 | V-16 | R-18 | M-2 | V-16 | R-18 | M-2 |
| BadNets | w/o CLIP-CFE | 0.920 | 0.926 | 0.928 | 0.921 | 0.928 | 0.928 | 0.923 | 0.928 | 0.928 | 0.921 | 0.928 | 0.929 |
| | w/ CLIP-CFE | 0.920 | 0.927 | 0.928 | 0.920 | 0.927 | 0.928 | 0.920 | 0.926 | 0.927 | 0.920 | 0.927 | 0.928 |
| Blended | w/o CLIP-CFE | 0.921 | 0.927 | 0.930 | 0.921 | 0.928 | 0.929 | 0.919 | 0.928 | 0.928 | 0.920 | 0.926 | 0.929 |
| | w/ CLIP-CFE | 0.921 | 0.926 | 0.929 | 0.921 | 0.927 | 0.929 | 0.920 | 0.927 | 0.929 | 0.921 | 0.927 | 0.929 |
| CLIP-UAP | w/o CLIP-CFE | 0.922 | 0.928 | 0.927 | 0.921 | 0.928 | 0.928 | 0.920 | 0.928 | 0.928 | 0.920 | 0.930 | 0.928 |
| | w/ CLIP-CFE | 0.922 | 0.928 | 0.929 | 0.922 | 0.928 | 0.929 | 0.920 | 0.927 | 0.929 | 0.922 | 0.928 | 0.928 |
| CLIP-CFA | w/o CLIP-CFE | 0.922 | 0.928 | 0.928 | 0.921 | 0.927 | 0.927 | 0.920 | 0.928 | 0.928 | 0.921 | 0.929 | 0.928 |
| | w/ CLIP-CFE | 0.922 | 0.929 | 0.927 | 0.920 | 0.927 | 0.928 | 0.921 | 0.929 | 0.929 | 0.921 | 0.929 | 0.928 |

and the corresponding poisoning image $x'_i = \mathcal{T}(x_i, t)$, the PSNR is formulated as:

$$
\begin{aligned}
\text{PSNR} &= \frac{1}{M} \sum_{i=1}^{M} \text{PSNR}_i(x_i, x'_i), \\
\text{where} \quad \text{PSNR}_i(x_i, x'_i) &= 10 \log_{10} \left( 255^2 / \text{MSE}(x_i, x'_i) \right), \\
\text{MSE}(f, g) &= \frac{1}{HW} \sum_{i=1}^{H} \sum_{j=1}^{W} (f_{ij} - g_{ij})^2,
\end{aligned}
\tag{16}
$$

$H$ and $W$ are height and width of the image, respectively. Larger PSNR means larger similarity between clean images and the corresponding poisoning images, therefore larges stealthiness of backdoor attacks.

**Structural Similarity Index (SSIM)** Wang et al. (2004). Similar to PSNR, SSIM is another metrics to represent the stealthiness of backdoor attacks, which is formulated as:

$$
\begin{aligned}
\text{SSIM} &= \frac{1}{M} \sum_{i=1}^{M} \text{SSIM}_i(x_i, x'_i), \\
\text{where} \quad \text{SSIM}_i(x_i, x'_i) &= l(x_i, x'_i) \cdot c(x_i, x'_i) \cdot s(x_i, x'_i), \\
\begin{cases}
l(f, g) = \frac{2\mu_f \mu_g + C_1}{\mu_f^2 + \mu_g^2 + C_1} \\
c(f, g) = \frac{2\sigma_f \sigma_g + C_2}{\sigma_f^2 + \sigma_g^2 + C_2} \\
s(f, g) = \frac{\sigma_{fg} + C_3}{\sigma_f \sigma_g + C_3}
\end{cases} &,
\end{aligned}
\tag{17}
$$

where $\mu$ and $\sigma$ are mean and variance of image, respectively. Similarly, Larger SSIM means larger similarity between clean images and the corresponding poisoning images, therefore larges stealthiness of backdoor attacks.

## A.8   RQ3: ARE PROPOSED TECHNOLOGIES STEALTHY FOR VICTIMS?

**Qualitative and Quantitative Results.** Fig. 8 showcases examples of poisoning images generated by different attacks on the ImageNet-50[4] dataset. While our CLIP-UAP and CLIP-CFA may not achieve the highest stealthiness in terms of SSIM (as indicated in Table 2), the poisoning images generated by our methods appear more natural to human inspection compared to the baseline attacks. Additionally, incorporating CLIP-CFE has minimal impact on both PSNR and the natural appearance of the images, while achieving higher stealthiness in terms of SSIM.

**Stealth in attack defense methods.** Attack stealthiness needs to be evaluated through algorithm. **(i)** As depicted in Figure 9, our preliminary evaluation conducted on the VGG-16 and CIFAR-100

---

[4]As shown in Appendix A.7, CIFAR-10 and CIFAR-100 have low resolution, which makes unclear visualizations. Therefore, we show the results on the ImageNet-50 dataset in this section.

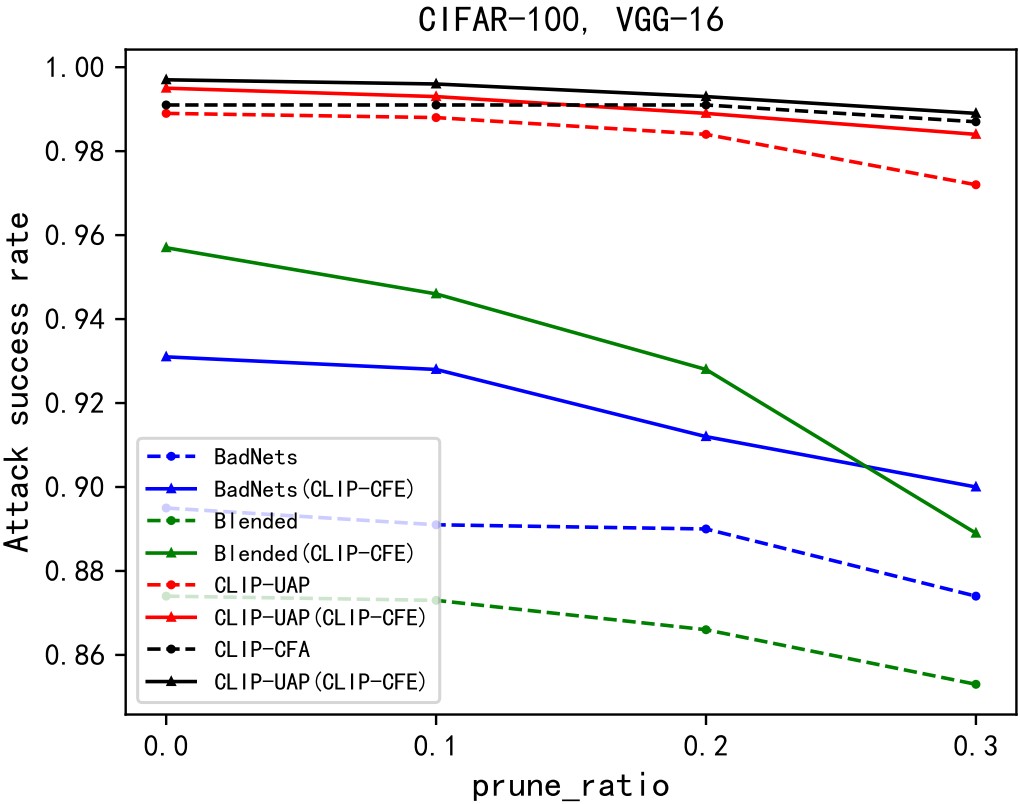

Figure 9: Defense results of pruning Liu et al. (2018) on the number-constrained backdoor attack of the CIFAR-100 dataset and VGG-16 model, where the number of clean samples owned by the defender is 50 and the poisoned rate is 2%.

datasets vividly demonstrates that our proposed method can attack against pruning-based defense method Liu et al. (2018) more effectively than other attack methods. **(ii)** As depicted in Figure 10, our preliminary evaluation conducted on the ResNet-18 model and CIFAR-100 datasets vividly demonstrates that our proposed method can attack against another pruning-based defense method Neural cleanse Wang et al. (2019a) more effectively than other attack methods. **(iii)** Similarly, as depicted in Figure 11, our preliminary evaluation conducted on the ResNet-18 and CIFAR-100 datasets vividly demonstrates that our proposed method can attack against tune-based defense method FST Min et al. (2023) more effectively than other attack methods.

## A.9 EXPERIMENTS ON THE CIFAR-10 DATASET

### A.9.1 RQ1: ARE PROPOSED TECHNOLOGIES EFFECTIVE ON DIFFERENT BACKDOOR ATTACKS.

In this section, we utilize our proposed technologies to attack different target models on the CIFAR-10 dataset. Our objective is to verify the effectiveness of the attack and calculate the ASR for each target model. The baseline attack methods, BadNets Gu et al. (2019) and Blended Chen et al. (2017) were introduced in Sec. A.2.1. The attack performance of the number-constrained, clean-label single-class (class-constrained), dirty-label single-class (class-constrained), and out-of-the-domain (domain-constrained) backdoor attacks on CIFAR-10 dataset are reflected in Fig. 12, 13, 14, and 15, respectively.

**CLIP-based Poisoning Feature Augmentation Is More Effective Than Previous Attack Methods.** Our proposed CLIP-UAP and CLIP-CFA methods outperform the BadNets Gu et al. (2019) and Blended Chen et al. (2017) baseline methods in terms of ASR under the most attacks and datasets.

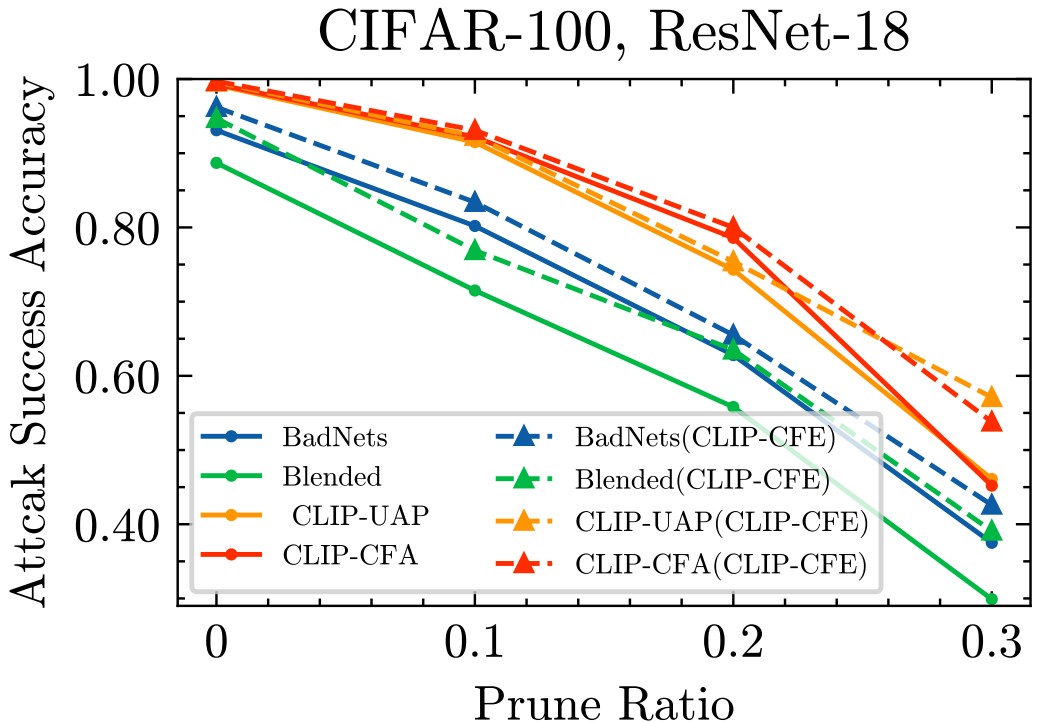

Figure 10: Defense results of Neural cleanse Wang et al. (2019a) on the number-constrained backdoor attack of the CIFAR-100 dataset and ResNet-18 model, where the number of clean samples owned by the defender is 1500, and the poisoned rate is 2%.

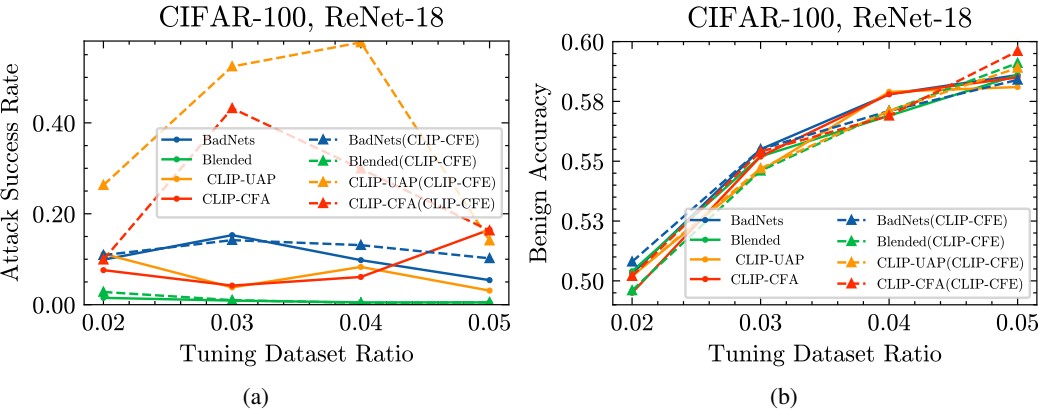

Figure 11: Defense results of FST Min et al. (2023) on the number-constrained backdoor attack of the CIFAR-100 dataset and ResNet-18 model, where the number of clean samples owned by the defender is from 1000 to 2500 and the poisoned rate is 2%.

This confirms that the proposed poisoning feature augmentation generates more efficient triggers than other methods.

**CLIP-based Clean Feature Suppression Is Useful For Different Attack Methods.** Our proposed CLIP-CFE method improves the poisoning effectiveness on most cases compared to the baseline without CLIP-based Clean Feature Erasing. Only on small cases the BadNets and CLIP-UAP slightly outperform the corresponding methods with CFE.

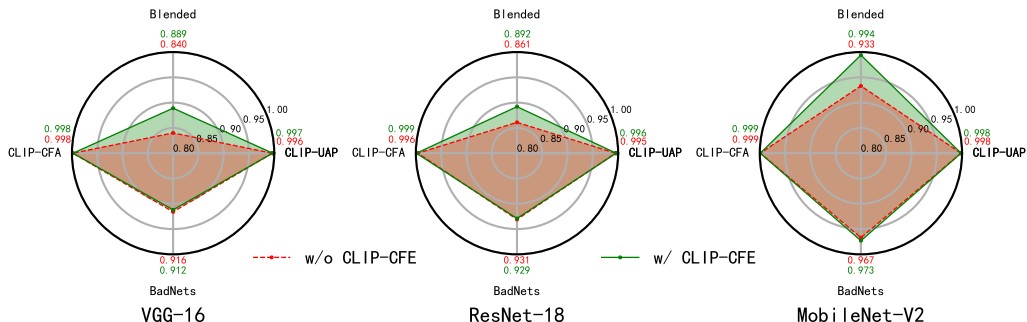

Figure 12: Attack success rate (ASR) of the number-constrained backdoor attacks on the CIFAR-10 dataset. The red points represents w/o CLIP-based Clean Feature Erasing (CFE), while the green points represents w/ CLIP-based Clean Feature Erasing (CFE). The experiment is repeated 5 times, and the results were computed as the mean of five different runs.

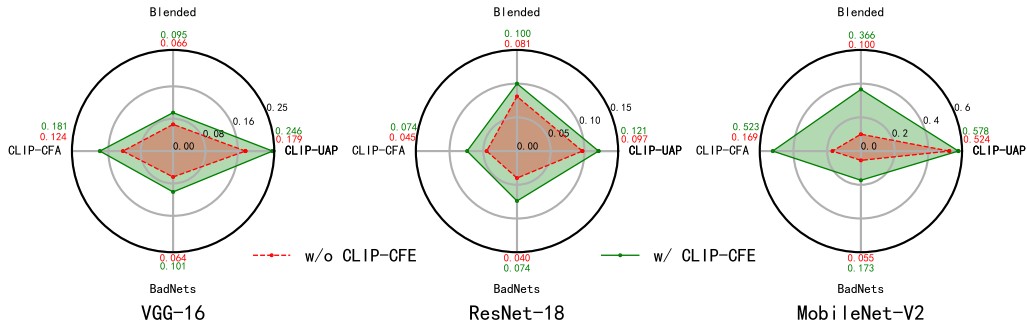

Figure 13: The attack success rate (ASR) of the class-constrained backdoor attacks (the access category $Y'$ is set to $\{0\}$) on the CIFAR-10 dataset. The red points represents w/o CLIP-based Clean Feature Erasing (CFE), while the green points represents w/ CLIP-based Clean Feature Erasing (CFE). The experiment is repeated 5 times, and the results were computed as the mean of five different runs.

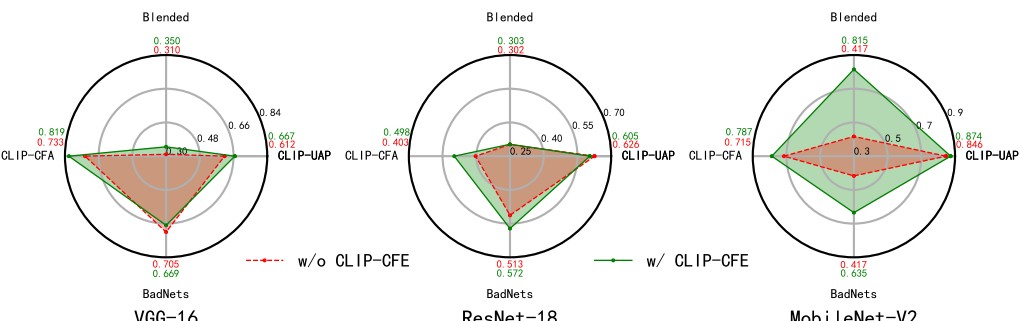

Figure 14: The attack success rate (ASR) of the class-constrained backdoor attacks (the access category $Y'$ is set to $\{1\}$) on the CIFAR-10 dataset. The red points represents w/o CLIP-based Clean Feature Erasing (CFE), while the green points represents w/ CLIP-based Clean Feature Erasing (CFE). The experiment is repeated 5 times, and the results were computed as the mean of five different runs.

### A.9.2 RQ2: ARE PROPOSED THREE TECHNOLOGIES HARMLESS FOR BENIGN ACCURACY.

Table 3 illustrates that our proposed CLIP-UAP and CLIP-CFA methods have similar or even better average Benign Accuracy (BA) compared to the baseline methods BadNets Gu et al. (2019) and Blended Chen et al. (2017). Additionally, our proposed CLIP-CFE method has no negative effect on

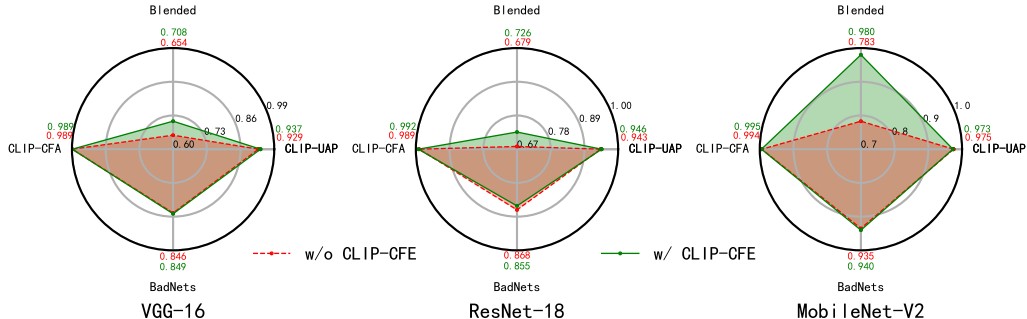

Figure 15: The attack success rate (ASR) of the domain-constrained backdoor attacks (Domain rate is set to 0) on the CIFAR-10 dataset. The red points represents w/o CLIP-based Clean Feature Erasing (CFE), while the green points represents w/ CLIP-based Clean Feature Erasing (CFE). The experiment is repeated 5 times, and the results were computed as the mean of five different runs.

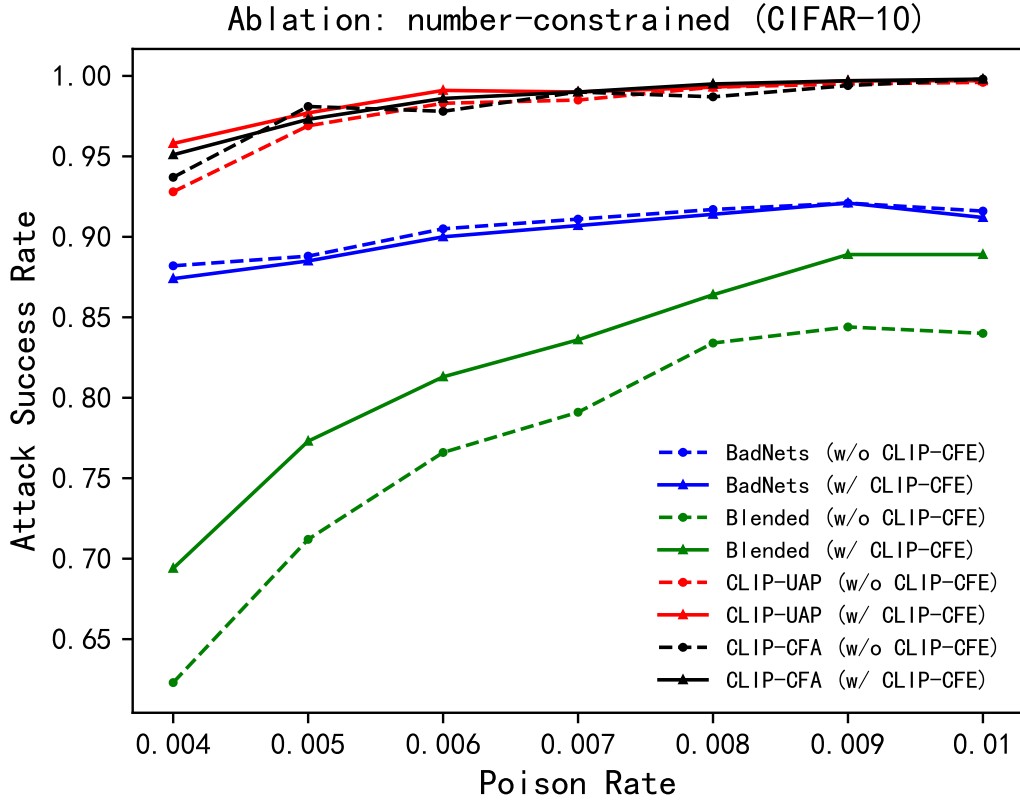

Figure 16: The attack success rate on the CIFAR-10 dataset at different poisoning rates. All results were computed as the mean of five different runs.

BA, confirming that our technologies are harmless for benign accuracy under various settings and different backdoor attacks.

### A.9.3 RQ3: ARE PROPOSED TECHNOLOGIES EFFECTIVE FOR DIFFERENT POISONING SETTINGS.

**Ablation of Different Poison Rates on The Number-constrained Backdoor Attacks.** We conducted ablation studies to verify the effectiveness of the proposed methods in reducing the number of poisoning samples (poisoning rates) on the number-constrained backdoor attacks. The results in Fig. 16 illustrate that: i) The attack success rate increases with the increase of poisoning rate

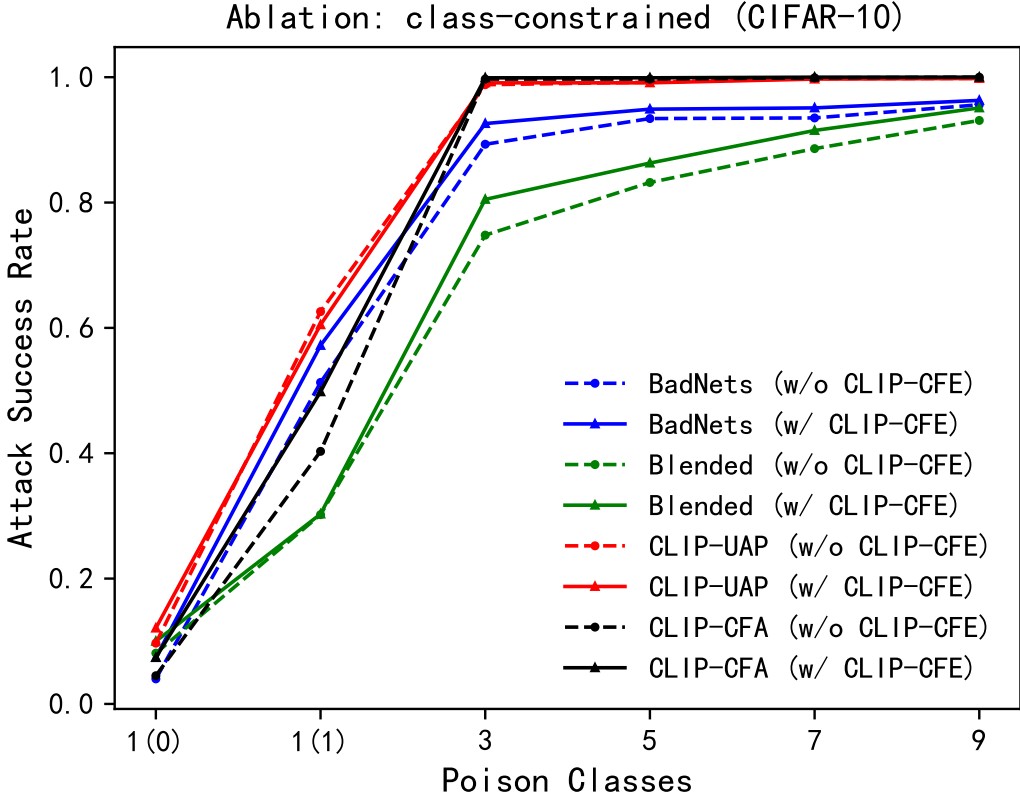

Figure 17: The attack success rate on the CIFAR-10 dataset at different accessible class of poisoning samples. All results were computed as the mean of five different runs.

for different attacks; ii) Our proposed CLIP-UAP and CLIP-CFA methods outperform the BadNets Gu et al. (2019) and Blended Chen et al. (2017); iii) The proposed CLIP-CFE further improves the poisoning effectiveness upon the different triggers.

**Ablation of Different Poison Classes on The Class-constrained Backdoor Attacks.** In this section, we conducted ablation studies to verify the effectiveness of the proposed methods in increasing the number of poisoning classes on the class-constrained backdoor attacks. The results in Fig. 17 illustrate that: i) The attack success rate increases with the increase of poisoning classes for different attacks; ii) The attack success rate of clean-label single-class attack is lower than that of dirty-label single-class attacks; iii) Our proposed CLIP-UAP and CLIP-CFA methods outperform the BadNets Gu et al. (2019) and Blended Chen et al. (2017) methods; iv) The proposed CLIP-CFE method further improves the poisoning effectiveness with different triggers.

**Ablation of different domain rates on the domain-constrained backdoor attacks.** In this section, we conducted ablation studies to verify the effectiveness of the proposed methods in increasing the domain rate on the domain-constrained backdoor attacks. The results in Fig. 18 illustrate that: i) The attack success rate increases with the increase of the domain rates for different attacks; ii) Our proposed CLIP-UAP and CLIP-CFA methods outperform the BadNets Gu et al. (2019) and Blended Chen et al. (2017) methods; iii) The proposed CLIP-CFE method further improves the poisoning effectiveness with different triggers.

## A.10   EXPERIMENTS ON THE IMAGENET-50 DATASET

### A.10.1   RQ1: ARE PROPOSED TECHNOLOGIES EFFECTIVE ON DIFFERENT BACKDOOR ATTACKS.

In this section, we utilized our proposed technologies to attack different target models on the ImageNet-50 dataset and calculate the ASR for each target model to verify the attack effective-

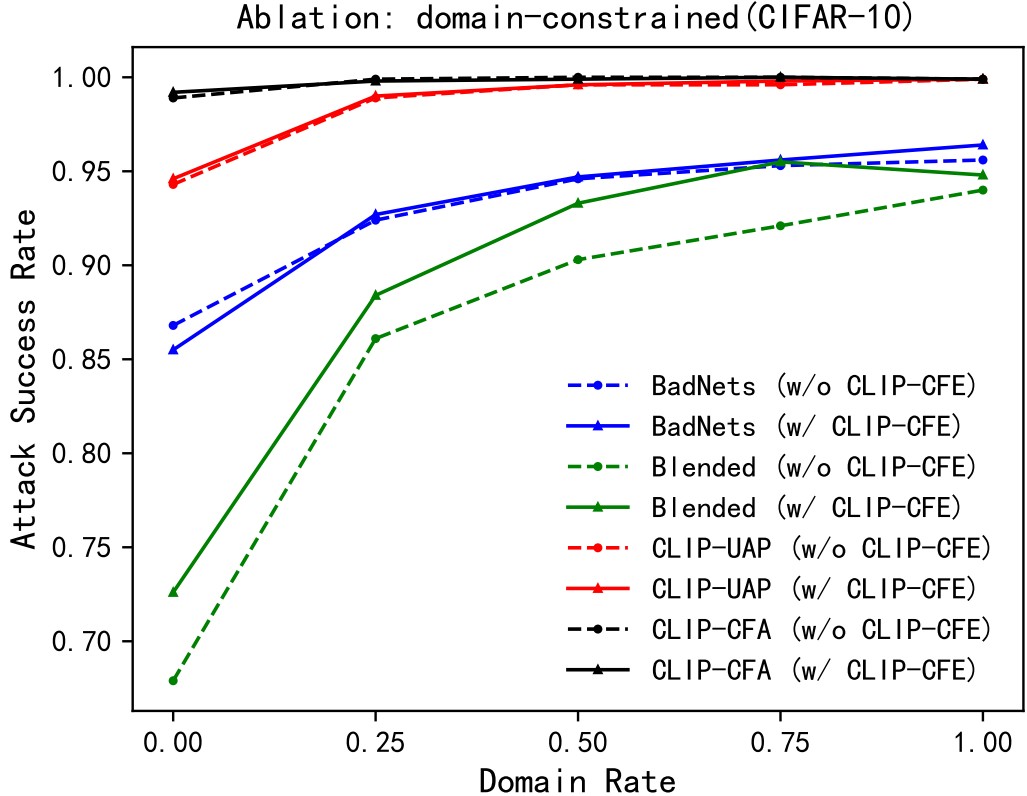

Figure 18: The attack success rate on the CIFAR-10 dataset at different proportions of in-domain samples. All results were computed as the mean of five different runs.

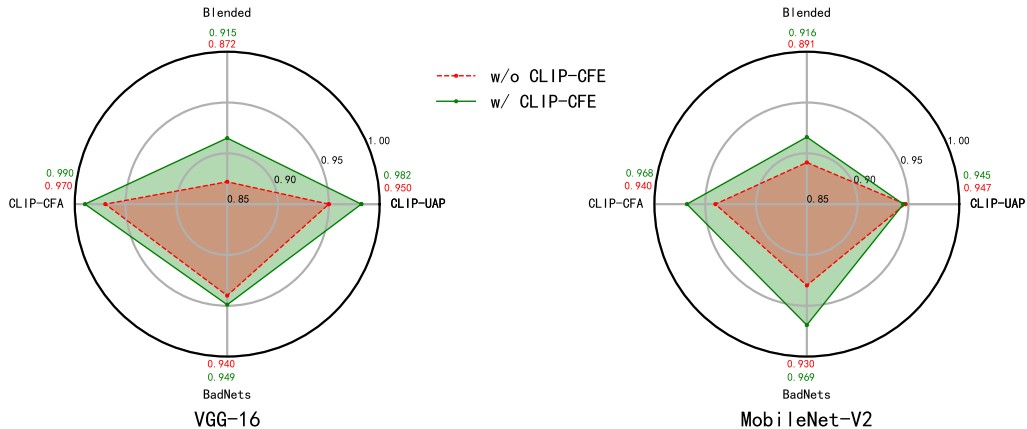

Figure 19: Attack success rate (ASR) of the number-constrained backdoor attacks on the ImageNet-50 dataset. The red points represents w/o CLIP-based Clean Feature Erasing (CFE), while the green points represents w/ CLIP-based Clean Feature Erasing (CFE). The experiment is repeated 5 times, and the results were computed as the mean of five different runs.

ness. The baseline attack methods, BadNets Gu et al. (2019)and Blended Chen et al. (2017) were introduced in Sec. A.2.1. Fig. 19, 20, 21, and 22 reflect the attack performance of the number-constrained, clean-label single-class (class-constrained), dirty-label single-class (class-constrained), and out-of-the domain (domain-constrained) backdoor attacks on the ImageNet-50 dataset, respectively.

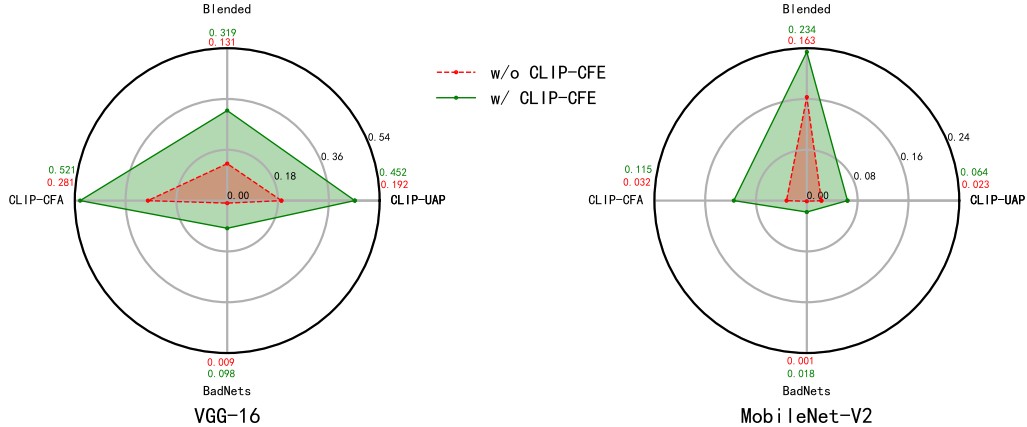

Figure 20: The attack success rate (ASR) of the class-constrained backdoor attacks (the access category $Y'$ is set to $\{0\}$) on the ImageNet-50 dataset. The red points represents w/o CLIP-based Clean Feature Erasing (CFE), while the green points represents w/ CLIP-based Clean Feature Erasing (CFE). The experiment is repeated 5 times, and the results were computed as the mean of five different runs.

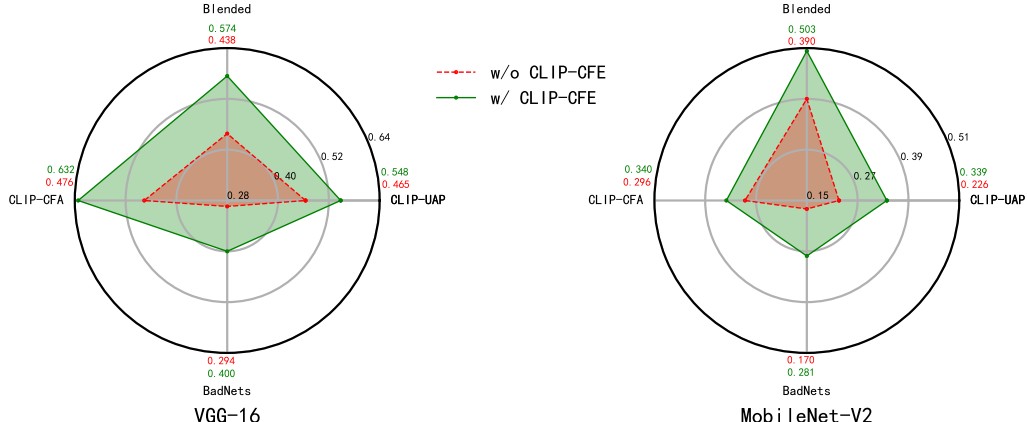

Figure 21: The attack success rate (ASR) of the class-constrained backdoor attacks (the access category $Y'$ is set to $\{1\}$) on the ImageNet-50 dataset. The red points represents w/o CLIP-based Clean Feature Erasing (CFE), while the green points represents w/ CLIP-based Clean Feature Erasing (CFE). The experiment is repeated 5 times, and the results were computed as the mean of five different runs.

**CLIP-based Poisoning Feature Augmentation Is More Effective Than Previous Attack Methods.** Our proposed CLIP-UAP and CLIP-CFA methods outperform the BadNets Gu et al. (2019) and Blended Chen et al. (2017) baseline methods in terms of consistency under different attacks and datasets. This confirms that the proposed poisoning feature augmentation generates more efficient triggers than other methods.

**CLIP-based Clean Feature Suppression Is Useful For Different Attack Methods.** Our proposed CLIP-CFE method improves the poisoning effectiveness on most cases compared to the baseline without CLIP-based Clean Feature Erasing. Only on the MobileNet-V2 results of the number-constrained backdoor attacks (right part of the Fig. 19), CLIP-UAP slightly outperform the corresponding methods with CFE.

A.10.2   RQ2: ARE PROPOSED THREE TECHNOLOGIES HARMLESS FOR BENIGN ACCURACY.

Table 4 illustrates that our proposed CLIP-UAP and CLIP-CFA methods have similar or even better average Benign Accuracy (BA) compared to the baseline methods BadNets Gu et al. (2019) and

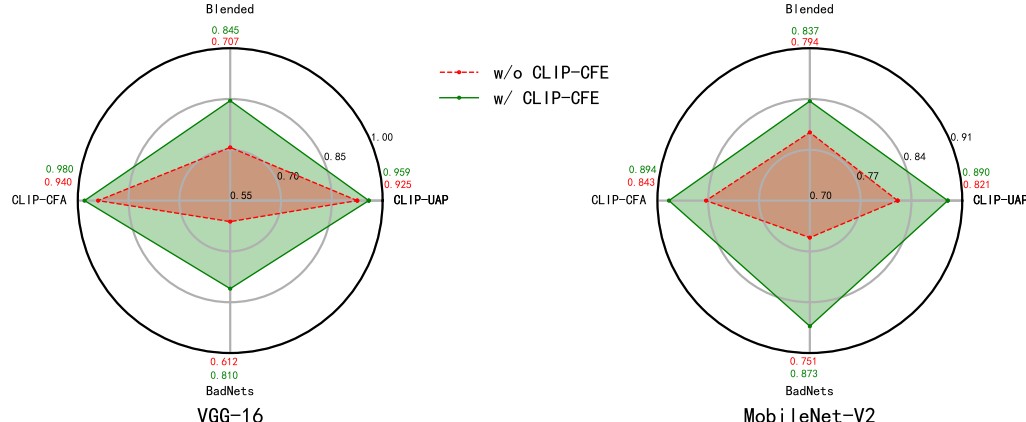

Figure 22: The attack success rate (ASR) of the domain-constrained backdoor attacks (Domain rate is set to 0) on the ImageNet-50 dataset. The red points represents w/o CLIP-based Clean Feature Erasing (CFE), while the green points represents w/ CLIP-based Clean Feature Erasing (CFE). The experiment is repeated 5 times, and the results were computed as the mean of five different runs.

Table 4: The Benign Accuracy (BA) on the ImageNet-50 dataset. All results are computed the mean by 5 different run.

| Trigger | Clean Feature Suppression | Backdoor Attacks | | | | | | | |
|---|---|---|---|---|---|---|---|---|---|
| | | Number Constrained | | Class Constrained ($Y' = \{0\}$) | | Class Constrained ($Y' = \{1\}$) | | Domain Constrained | |
| | | V-16 | M-2 | V-16 | M-2 | V-16 | M-2 | V-16 | M-2 |
| BadNets | w/o CLIP-CFE | 0.784 | 0.731 | 0.788 | 0.731 | 0.788 | 0.733 | 0.783 | 0.735 |
| | w/ CLIP-CFE | 0.787 | 0.733 | 0.788 | 0.735 | 0.790 | 0.734 | 0.788 | 0.735 |
| Blended | w/o CLIP-CFE | 0.789 | 0.733 | 0.788 | 0.736 | 0.787 | 0.727 | 0.788 | 0.731 |
| | w/ CLIP-CFE | 0.790 | 0.731 | 0.787 | 0.734 | 0.788 | 0.731 | 0.786 | 0.728 |
| CLIP-UAP | w/o CLIP-CFE | 0.788 | 0.729 | 0.788 | 0.726 | 0.786 | 0.728 | 0.789 | 0.733 |
| | w/ CLIP-CFE | 0.785 | 0.732 | 0.786 | 0.729 | 0.786 | 0.734 | 0.791 | 0.730 |
| CLIP-CFA | w/o CLIP-CFE | 0.784 | 0.730 | 0.789 | 0.732 | 0.785 | 0.729 | 0.787 | 0.735 |
| | w/ CLIP-CFE | 0.787 | 0.732 | 0.786 | 0.734 | 0.789 | 0.735 | 0.784 | 0.732 |

Blended Chen et al. (2017). Additionally, our proposed CLIP-CFE method has no negative effect on BA, confirming that our technologies are harmless for benign accuracy under various settings and different backdoor attacks.

## A.11 EXPERIMENTS ON MORE COMPLEX CONSTRAINTS IN DATA-CONSTRAINT BACKDOOR ATTACKS

Previous sections have primarily focused on specific sub-variants of number, class, and domain constraints, which might not comprehensively represent all real-world limitations. This section delves into more intricate constraints. Specifically, we investigate two additional configurations:

**Config A**: Poisoning rate = 0.01 (P=500), poisoning classes=1 (1), and domain rate=0.5.

**Config B**: Poisoning rate = 0.01 (P=500), poisoning classes=3, and domain rate=0.25.

The experiments are performed using the VGG-16 model and the CIFAR-100 dataset. As depicted in Fig. 23, our methods and conclusions are demonstrated to be equally viable in scenarios involving more complex data constraints.

## A.12 EXPERIMENTS ON DOMAINS DIFFERENT FROM CLIP'S TRAINING DOMAIN

In this section, we verify the performance of the proposed methods on domain drastically differs from CLIP's training set. Typically, we select the commonly used dataset UCM Yang & Newsam (2010) in the field of remote sensing.

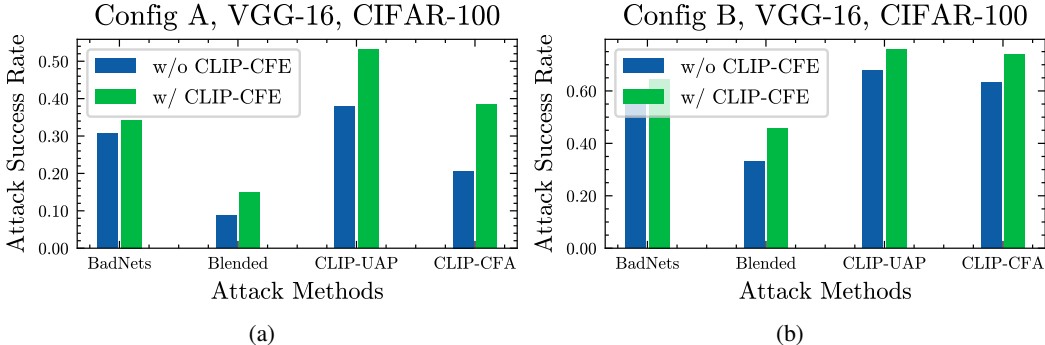

Figure 23: The Attack Success Rate (ASR) on more complex data-constraint attacks. (a) **Config A:** Poisoning rate = 0.01 (P=500), poisoning classes=1 (1), and domain rate=0.5. (b) **Config B:** Poisoning rate = 0.01 (P=500), poisoning classes=3, and domain rate=0.25.

Table 5: The Attack Success Rate (ASR) on the VGG-16 model and UCM Yang & Newsam (2010) dataset. All results are computed the mean by 5 different run. In the class-Constrained Scenario, the poisoning classes is set to 1, and we choose the dirty-label setting.

| Trigger | Clean Feature Suppression | Backdoor Attacks | | | | | |
|---|---|---|---|---|---|---|---|
| | | Number Constrained | | | Class Constrained | | |
| | | CLIP ($\epsilon$=8/255) | CLIP ($\epsilon$=16/255) | Satellite ($\epsilon$=8/255) | CLIP ($\epsilon$=8/255) | CLIP ($\epsilon$=16/255) | Satellite ($\epsilon$=8/255) |
| BadNets | w/o CLIP-CFE | 0.916 | 0.916 | 0.916 | 0.692 | 0.692 | 0.692 |
| | w/ CLIP-CFE | 0.921 | 0.944 | 0.952 | 0.661 | 0.791 | 0.837 |
| Blended | w/o CLIP-CFE | 0.798 | 0.798 | 0.798 | 0.329 | 0.329 | 0.329 |
| | w/ CLIP-CFE | 0.805 | 0.882 | 0.904 | 0.338 | 0.474 | 0.526 |
| CLIP-UAP | w/o CLIP-CFE | 0.860 | 0.895 | 0.917 | 0.722 | 0.783 | 0.823 |
| | w/ CLIP-CFE | 0.962 | 0.981 | 0.990 | 0.719 | 0.824 | 0.865 |
| CLIP-CFA | w/o CLIP-CFE | 0.898 | 0.924 | 0.937 | 0.585 | 0.713 | 0.735 |
| | w/ CLIP-CFE | 0.961 | 0.980 | 0.989 | 0.717 | 0.805 | 0.833 |

**UCM Dataset.** The UCM contains 100 images in each of the 21 categories for a total of 2100 images. Each image has a size of $256 \times 256$ and a spatial resolution of 0.3 m/pixel. The images are captured in the optical spectrum and represented in the RGB domain. The data are extracted from aerial ortho imagery from the U.S. Geological Survey (USGS) National Map. The categories in this dataset are: agricultural, airplane, baseball diamond, beach, buildings, chaparral, dense residential, forest, freeway, golf course, harbor, intersection, medium-density residential, mobile home park, overpass, parking lot, river, runway, sparse residential, storage tanks, and tennis courts.

**Experiments.** Similar to Dräger et al. (2023), we randomly select 1050 samples to form the training set, while the remaining samples make up the test set. The attack-target class $k$ is set to "agricultural". The poisoning rate is set to 0.05 in all experiments. As illustrated in Table 5, we illustrate the Attack Success Rate of number-Constrained and class-Constrained backdoor attacks on UCM dataset. The results indicate a performance decline on the UCM dataset. However, as seen in Sec. A.11.1, by adjusting the constraints ($\epsilon$) of the optimized noise, we observed an improved attack success rate in our methods compared to baseline methods. Additionally, replacing CLIP with the Satellite Arto et al. (2021) model, a large model fine-tuned on remote sensing images, further augmented the attack success rate. These outcomes demonstrate the adaptability of our methods to various domains by replacing CLIP with domain-specific pre-trained models.

## A.13 EXPERIMENTS ON FINE-GRAINED DATASETS

In this section, we rigorously assess the effectiveness of our proposed methodologies using fine-grained datasets, with a specific focus on the widely recognized Oxford-Flowers dataset within the domain of fine-grained image classification.

**Oxford-Flowers Dataset.** The Oxford-Flowers dataset comprises 102 categories of common UK flower images, totaling 8189 images, with each category containing 40 to 258 images. Our exper-

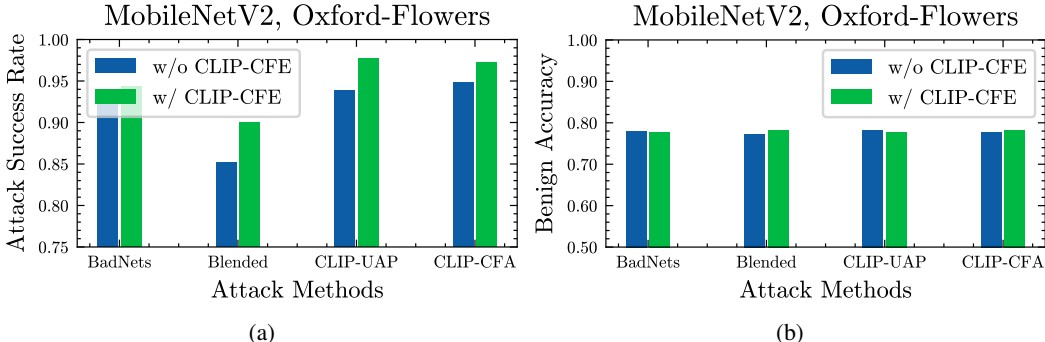

Figure 24: The Attack Success Rate (ASR) and Benign Accuracy (BA) on Oxford-Flowers dataset. Poisoning rate = 0.01 (P=307). All results were computed as the mean of five different runs.

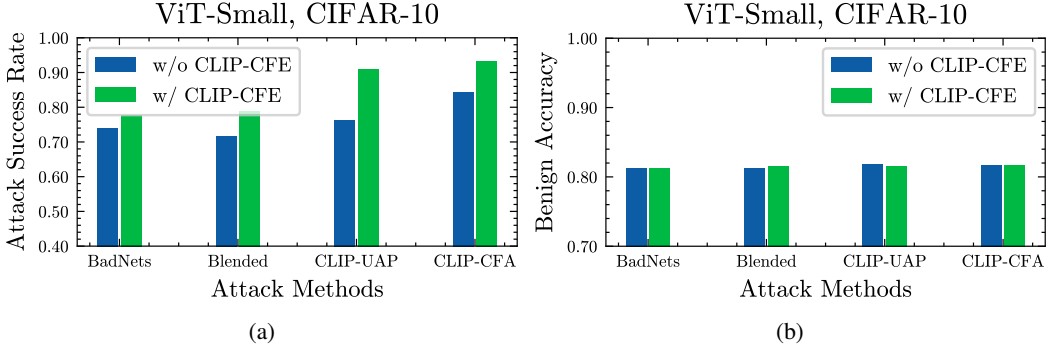

Figure 25: The Attack Success Rate (ASR) and Benign Accuracy (BA) on ViT-small architecture and CIFAR-10 dataset. Poisoning rate = 0.01 (P=500). All results were computed as the mean of five different runs.

imental setup involves employing 6140 images for training purposes, while the remaining samples constitute the test set.

**Experiments.** For our experiments, we set the attack-target class ($k$) to 0 and maintain a consistent poisoning rate of 0.05 (p=307) across all conducted trials. Illustrated in Figure 24 (a), we showcase the Attack Success Rate of number-Constrained backdoor attacks on the Oxford-Flowers dataset. Our findings conclusively demonstrate the superior efficacy of CLIP-based poisoning feature augmentation compared to prior attack methodologies. Additionally, CLIP-based Clean Feature Suppression emerges as a valuable strategy across diverse attack methods. Moreover, as depicted in Figure 24 (b), our results unequivocally indicate that our technologies maintain benign accuracy at par with baseline methods, even under varying settings and diverse backdoor attacks. This underlines the robustness and non-disruptive nature of our methodologies.

## A.14 Experiments on ViT Architecture

For our experiments, we set the attack-target class ($k$) to 0 and maintain a consistent poisoning rate of 0.05 (p=500) across all conducted trials. Illustrated in Figure 25 (a), we showcase the Attack Success Rate of number-Constrained backdoor attacks on the ViT-Small architecture and CIFAR-10 dataset. Our findings conclusively demonstrate the superior efficacy of CLIP-based poisoning feature augmentation compared to prior attack methodologies. Additionally, CLIP-based Clean Feature Suppression emerges as a valuable strategy across diverse attack methods. Moreover, as depicted in Figure 25 (b), our results unequivocally indicate that our technologies maintain benign accuracy at par with baseline methods, even under varying settings and diverse backdoor attacks. This underlines the robustness and non-disruptive nature of our methodologies.

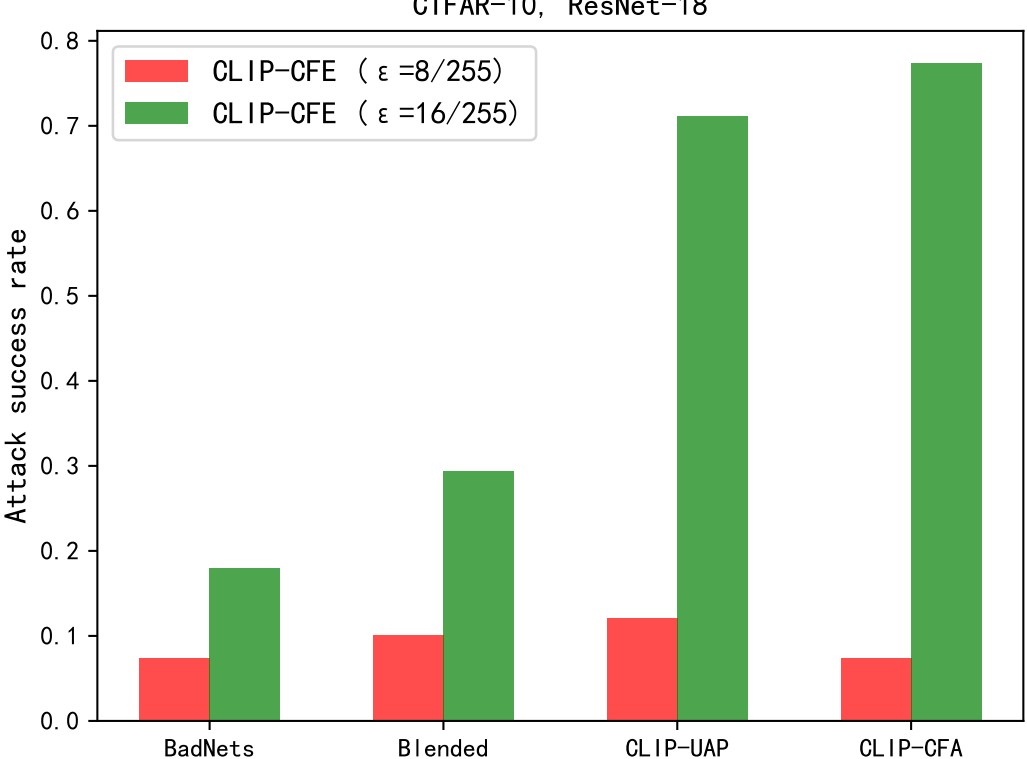

Figure 26: The attack success rate on the CIFAR-10 dataset at different $\epsilon$. All results are computed as the mean of five different runs.

## A.15 DISCUSSION

### A.15.1 PERFORMANCE DEGRADATION IN THE CLEAN-LABEL SINGLE-CLASS BACKDOOR ATTACK.

As depicted in Fig. 3 (b), Fig. 13, and Fig. 20, both the baseline and our attack methods exhibit poor Attack Success Rate (ASR) in the clean-label single-class backdoor attack. In this section, we aim to enhance the attack strength of our methods and devise a more efficient attack strategy for the clean-label single-class backdoor attack. In our optimization equations, namely Eq. 3, Eq. 13, and Eq. 10, we impose constraints on the optimized noise, denoted as $\delta_i$, $\delta_{\mathrm{uap}}$, and $\delta_{\mathrm{con}}$, respectively. These constraints are specified as $||\delta_i||_p \leq \epsilon$, $||\delta_{\mathrm{uap}}||_p \leq \epsilon$, and $||\delta_{\mathrm{con}}||_p \leq \epsilon$, where $|| \cdot ||_p$ represents the $L_p$ norm, and we set $\epsilon$ to $8/255$ to ensure the stealthiness of the backdoor attacks, as observed in our previous experiments. To bolster the attack strength and subsequently increase the ASR in the clean-label single-class backdoor attack, we investigate the impact of adjusting the constraint on $\delta_i$. As demonstrated in Fig. 26 and Fig. 27, significant (more than $500\%$) improvements are observed in the ASR of the clean-label single-class backdoor attack when we set the constraint on $\delta_i$ to $||\delta_i||_p \leq 16/255$. This finding validates the efficacy of our method in the clean-label single-class backdoor attack, albeit at the expense of compromising stealthiness. This sacrifice, which is common in previous backdoor attack methods Zeng et al. (2022), is a low-cost trade-off.

### A.15.2 DOMAIN-CONSTRAINED BACKDOOR ATTACKS ARE EASIER THAN CLASS-CONSTRAINED BACKDOOR ATTACKS.

Fig. 28 provides a visualization of the Attack Success Rate (ASR) achieved by different attack methods on the CIFAR-10 dataset in domain-constrained (domain rate set to 0) and dirty-label single-class backdoor attacks. While domain-constrained backdoor attacks impose stricter restrictions (assumptions that attackers have no access to any data in the training set), the ASR in domain-constrained backdoor attacks consistently surpasses that of dirty-label single-class backdoor attacks.

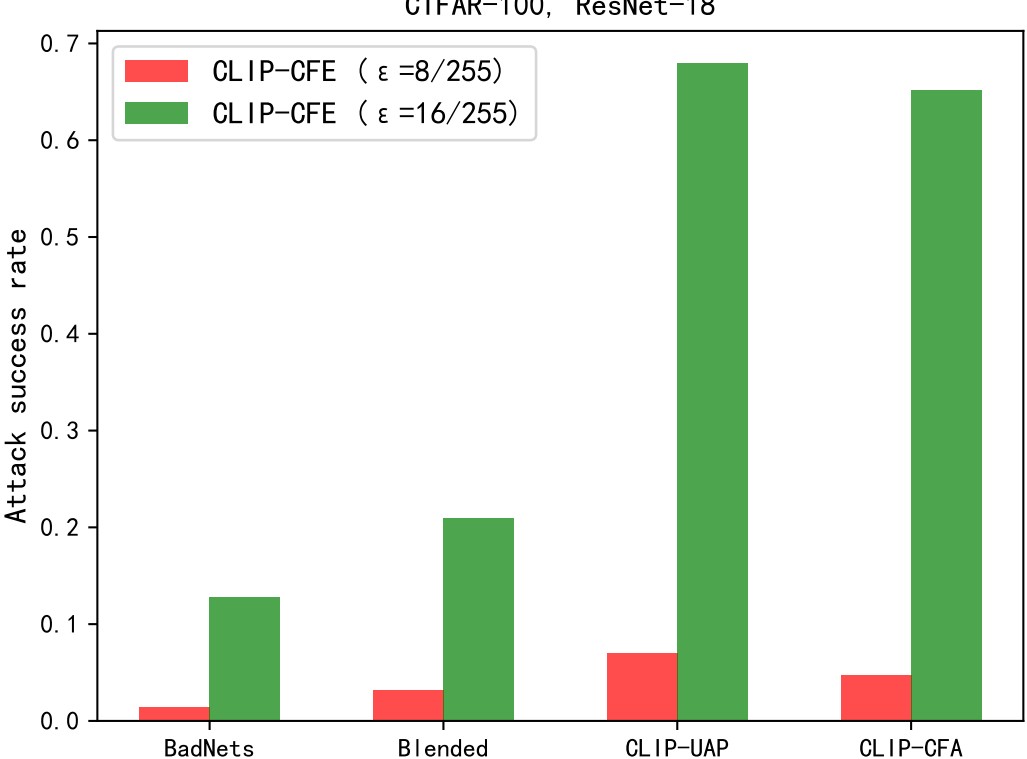

Figure 27: The attack success rate on the CIFAR-100 dataset at different $\epsilon$. All results are computed as the mean of five different runs.

This observation leads us to propose that the diversity of samples in the poisoning set is another crucial factor affecting attack efficiency. Consequently, we recommend that attackers fully consider the diversity of poisoning samples during the poisoning set generation phase.

### A.15.3 TIME CONSUMPTION OF DIFFERENT ATTACK METHODS.

We would like to highlight that in the case of BadNet and Blended, adding a pre-defined trigger into benign images to build poisoned images only needs to perform one time addition calculation, which incurs negligible time consumption. It's important to note, however, that these straightforward implementations fall short in terms of both poison efficiency and stealthiness in the context of backdoor attacks. Recent advancements in backdoor attack techniques have sought to enhance efficiency and covert effectiveness by introducing optimization-driven processes to define triggers. While these refinements do entail some additional time overhead, they significantly elevate the attack's efficacy. We undertook an evaluation of the time overhead associated with diverse attack methods using an A100 GPU. As outlined in Table 6, the time consumption exhibited by optimization-based methods grows linearly with the expansion of the poison sample count. Despite this, the overall time overhead remains remarkably modest, with the entirety of the process being accomplished within mere minutes.

### A.15.4 EXPLAINING THE DIFFERENCE BETWEEN DATA-CONSTRAINT BACKDOOR ATTACKS AND BACKDOOR ATTACKS OF FEDERATED LEARNING.

The landscape of backdoor attacks in the context of federated learning shares certain similarities with our proposed data-constraint backdoor attacks. Notably, both scenarios assume the utilization of training data originating from diverse sources. However, it's important to note that the threat model for backdoor attacks in federated learning adopts a distinct perspective. In this model, the attacker exercises complete control over one or several participants: (1) The attacker possesses authority over the local training data of any compromised participant. (2) It wields control over

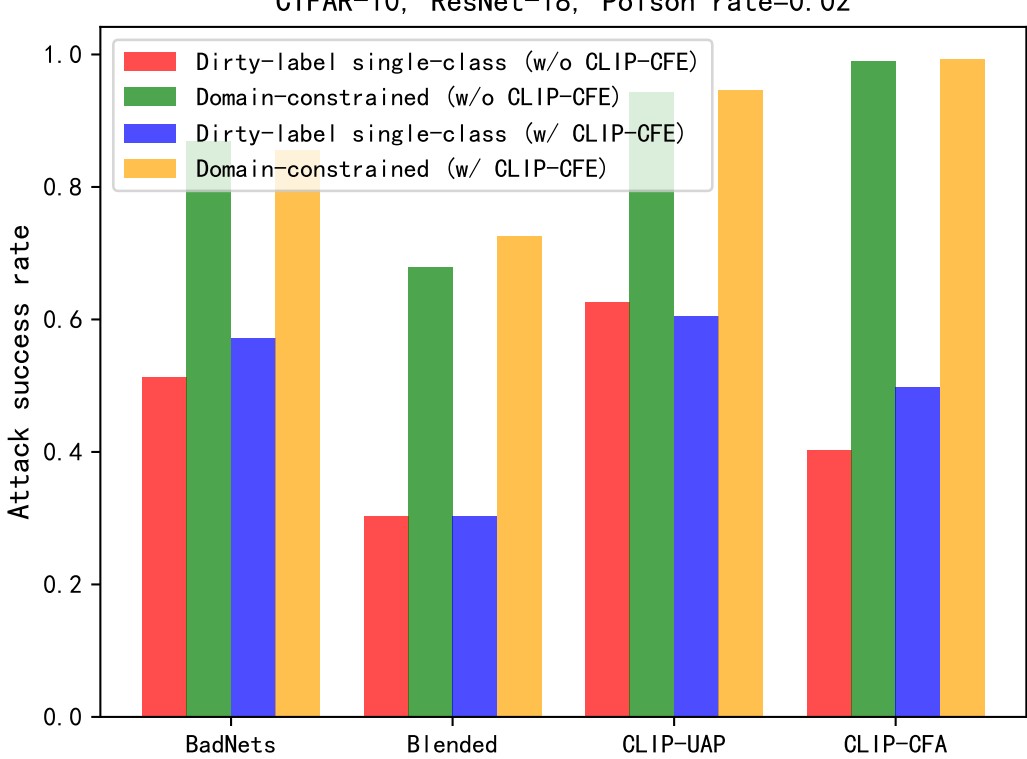

Figure 28: The attack success rate on the CIFAR-10 dataset at domain-constrained (domain rate is set to 0) and dirty-label single-class backdoor attacks. poisoning rate is set to 0.02 and all results are computed as the mean of five different runs.

the local training process and associated hyperparameters, including parameters like the number of training epochs and learning rate. (3) The attacker can manipulate the model's weights prior to submitting them for aggregation. (4) It retains the capability to dynamically adjust its local training strategy from one round to the next.

In contrast, our data-constraint backdoor attacks center exclusively on the attacker's capability to manipulate local training data. This introduces a heightened level of challenge for potential attackers. Furthermore, to the best of our knowledge, the existing backdoor attack strategies within the realm of federated learning primarily revolve around the number-constrained scenario. Notably, the class-constrained and domain-constrained scenarios have yet to be comprehensively explored within the community's discourse on this subject.

Table 6: The time overhead (min) of different attacks on a A100 GPU. All results are computed with the mean of 5 different runs.

| Number of Poisoned set | BadNet | BadNet +CLIP-CFE | UAP | UAP +CLIP-CFE | CLIP-UAP | CLIP-UAP +CLIP-CFE | CLIP-CFA | CLIP-CFA +CLIP-CFE |
|---|---|---|---|---|---|---|---|---|
| 500 | 0 | 1 | 0.7 | 1.6 | 1 | 2.1 | 0.9 | 1.9 |
| 1000 | 0 | 1.9 | 1.5 | 3.3 | 2 | 4 | 1.9 | 3.9 |
| 2000 | 0 | 3.7 | 2.9 | 6.5 | 4 | 7.9 | 3.9 | 7.8 |

## A.16 LIMITATIONS AND FUTURE WORKS

In this section, we discuss the limitations of our approach and outline potential future directions for backdoor learning research. **Performance Degradation in Clean-label Backdoor Attacks.** Clean-label backdoor attacks present a significant challenge Zhao et al. (2020). As shown in Fig.

3, previous methods exhibit a poor ASR, and our technologies show limited improvement in clean-label backdoor attacks when the poisoning rate is low. In future research, we will investigate the underlying reasons for this situation and explore more efficient attack methods specifically designed for clean-label backdoor attacks.

**Application Limitations.** Our technologies depend on the CLIP model that is pre-trained on natural images, which may limit their applicability to certain domains such as medical images or remote sensing. In such cases, a possible solution is to replace CLIP with a domain-specific pre-trained model, such as MedCLIP Madhawa & Carlomagno (2022) for medical images or Satellite Arto et al. (2021) for remote sensing, to adapt our methods to the target domain.

**Transfer to Other Domains.** The attack scenario we have defined is not limited to a specific domain and can be applied to other important applications, including backdoor attacks for malware detection, deepfake detection, and federated learning. In our future work, we plan to explore the design of realistic attack scenarios and efficient backdoor attacks specifically tailored for these applications.

