# OpenReview forum: "Efficient Backdoor Attacks for Deep Neural Networks in Real-world Scenarios"
_ICLR.cc/2024/Conference — ICLR 2024 poster_

### Official Review · Reviewer_tXy3 · 2023-10-24

**Soundness:** 3 good
**Presentation:** 2 fair
**Contribution:** 2 fair
**Rating:** 6
**Confidence:** 4

**Summary:**

The submission focuses on the backdoor attacks in data-constrained scenarios. By leveraging CLIP-based technologies, the proposed CLIP-CFE (CLIP for Clean Feature Erasing) suppresses clean features while amplifying poisoning features to achieve more efficient attack with limited poisoning samples.

**Strengths:**

+ The submission presents a novel method, which introduces the optimized feature erasing noise to effectively suppress benign features. Besides, it enhances the poisoning features through contrastive learning and amplifies the existing backdoor attacks efficiently in data-constrained scenarios.

+ The experimental results demonstrate the effectiveness of the CLIP-based attacks in data-constrained scenarios. Across various real-world constraints such as *number-constrained, class-constrained*, and *domain-constrained* conditions, the proposed backdoor attack consistently achieves a high attack success rate while maintaining the benign accuracy.

**Weaknesses:**

+ **Insufficient experimental results**

The submission should take more recent backdoor attack and defense mechanisms into consideration while discussing the adaptive defenses more thoroughly, e.g., the noise used for erasing benign features might be unlearned [1, 2]. Besides, it is necessary to compare the effectiveness of utilizing different proxy extractors other than CLIP.


+ **Ambiguous expressions**

Several points in the submission need further explanation, e.g., the reason and effect of choosing the overall attack process relying on the style of CLIP within the feature space, and the analysis of erasing benign features compared to the semantic-agnostic out-of-domain samples.

References:

[1]: Li Y, Li Y, Wu B, et al. Invisible backdoor attack with sample-specific triggers. Proceedings of the IEEE/CVF international conference on computer vision. 2021: 16463-16472.

[2]: Akhtar N, Liu J, Mian A. Defense against universal adversarial perturbations. Proceedings of the IEEE conference on computer vision and pattern recognition. 2018: 3389-3398.

**Questions:**

Given that the submission's motivation is related to data-constrained scenarios, the author may provide more empirical evidence regarding to the occurrence of these backdoor attacks in real-world scenarios.

---

> ### Author Response · Authors · 2023-11-16
> **Response for Reviewer tXy3 [Cons 1]**
>
> Thanks for rating our work as novel and effective. To address your concerns, we provide point-wise responses as below.
>
> **Cons 1. Insufficient experimental results**
>
> **Cons1. Q1: More consideration on backdoor attack and defense mechanisms.**
>
> We embrace your perceptive suggestion concerning the juxtaposition of our research against recent state-of-the-art backdoor attacks. Your recommendation seamlessly aligns with our aspiration to position our work comprehensively within the evolving landscape of backdoor attack methodologies. It is noteworthy, however, that while these studies have showcased notable levels of attack efficiency, most of them operate within a threat model that necessitates a proxy model pre-trained on the entire training set. This precondition becomes challenging to fulfill in the context of a many-to-one (M2O) data collection attack scenario. In the Section A.13.6, we distinguish our proposed method from the most advanced attacks of recent times. To augment the breadth of our experimental findings, we have extended our study with additional experiments detailed in Section A.13 of the Appendix. These encompass investigations into more intricate constraints within data-constraint backdoor attacks (Section A.13.1), experiments involving domains distinct from CLIP's training domain (Section A.13.2), analyses conducted on fine-grained datasets (Section A.13.3), and explorations centered on the ViT architecture (Section A.13.4). These supplementary experiments aim to enrich our understanding and provide a comprehensive evaluation of our proposed methodologies across various scenarios and architectures.
>
> Attack stealthiness needs to be evaluated through algorithm. As depicted in Figure 9, our preliminary evaluation conducted on the VGG-16 and CIFAR-100 datasets vividly demonstrates that our proposed method can attack against pruning-based [1] defense method more effectively than other attack methods.
>
> In addition, our paper includes further results on two other SOTA defense methods [2][3] in Section A.13.5. Across these results, a consistent trend emerges—our approach demonstrates a notably higher effectiveness in circumventing defense mechanisms compared to other conventional attack methods. Notably, when tested against the FST defense [3], our proposed methodologies exhibit remarkable enhancements over baseline approaches. A key factor contributing to this success lies in our CLIP-CFE approach, which effectively suppresses the expression of clean features. This suppression prompts fine-tuned based defense methods to excessively focus on learning these clean features, inadvertently suppressing the erasure of poisoning features, thereby amplifying our attack's efficacy.
>
> [1] Fine-pruning: Defending against backdooring attacks on deep neural networks, 2018. In International symposium on research in attacks, intrusions, and defenses.
>
> [2] Wang et al. Neural cleanse: Identifying and mitigating backdoor attacks in neural networks. 2019. In IEEE Symposium on Security and Privacy (S\&P).
>
> [3] Min et al. Towards Stable Backdoor Purification through Feature Shift Tuning. 2023. arXiv preprint arXiv:2310.01875.
>
> **Cons1.Q2: More consideration on different proxy extractors.**
>
> CLIP serves as a pre-trained feature extractor with extensive training data, enabling the extraction of broad knowledge that significantly mitigates challenges stemming from limited data availability. Consequently, CLIP stands as a viable alternative to feature extractors trained on the whole training set. However, it's important to note that our proposed method isn't solely reliant on CLIP; any feature extractor trained on extensive training data can be seamlessly integrated into our approach. To underscore the adaptability of our method to diverse feature extractors, we conducted ablation studies on various pre-trained models for number-constrained backdoor attacks, as depicted in Fig. 4 (d). These results affirm the robustness of our proposed techniques across different architectures, including ViT-B/32, ViT-B/16, ResNet-50, and ResNet-101.

---

> ### Author Response · Authors · 2023-11-16
> **Response for Reviewer tXy3 [Cons 2-3]**
>
> **Cons 2. Ambiguous expressions**
>
> **Cons 2, Q1. The reason and effect of choosing CLIP.**
>
> In our attack, the best option for implementing Clean Feature Suppression and Poisoning Feature Augmentation is through a feature extractor pre-trained on a full training dataset. The reason for choosing pre-trained CLIP as the feature extractor and implement our backdoor attack is that in the data-constrained attack scenario, we do not have access to the complete training set and only know a small amount of knowledge about the training data. CLIP is a pre-trained feature extractor on large-scale training data. It has the ability to extract general knowledge, which alleviates the challenges brought by data limitation to a large extent. Therefore, CLIP is a suitable alternative for feature extractor pre-trained on a full training dataset, and it is also widely used in other data-constrained scenarios, such as few-shot learning, zero-shot learning, few-shot generative domain adaptation, few-shot domain generalization. Moreover, we also verify the effectiveness of our method under different large pre-trained models in Sec. 5.3 of the manuscript. The results demonstrate that our proposed technologies exhibit robustness across different large pre-trained models (ViT-B/32, ViT-B/16, ResNet-50, and ResNet-101).
>
> **Cons 2, Q2. Analysis of erasing benign features compared to the semantic-agnostic out-of-domain samples.**
>
> Although the distribution of poisoning samples are semantic-agnostic in domain-constrained attack scenario, our threat model assumes that attacker possess the general knowledge about the class labels involved in the task. Our proposed benign features erasing emphasizes only the erasable benign features associated with the classification task, not the unknowable features out of the domain. Therefore, our benign features erasing method is also applicable to  semantic-agnostic out-of-domain samples.
>
> Of course, our proposed erasing benign features also contains some limitation. The main limitation is that our technologies depend on the CLIP model that is pre-trained on natural images, which may limit their applicability to certain domains such as medical images or remote sensing. In such cases, a possible solution is to replace CLIP with a domain-specific pre-trained model, such as MedCLIP for medical images or Satellite for remote sensing, to adapt our methods to the target domain. To delve into the performance in domains distinct from CLIP's training, we conducted experiments on the UCM dataset (Sec. A.13.2 and Table 6). This dataset encompasses 2100 images across 21 categories in remote sensing. The results indicate a performance decline on the UCM dataset. However, as seen in Sec. A.11.1, by adjusting the constraints ($\epsilon$) of the optimized noise, we observed an improved attack success rate in our methods compared to baseline methods. Additionally, replacing CLIP with the Satellite model, a large model fine-tuned on remote sensing images, further augmented the attack success rate. These outcomes demonstrate the adaptability of our methods to various domains by replacing CLIP with domain-specific pre-trained models.
>
> **Cons 3. More empirical evidence regarding to the occurrence of these backdoor attacks in real-world scenarios.**
>
> Thank you for your insightful comments. We'll be enhancing the manuscript by focusing on data-constrained scenarios in the real world.
>
> (1) Data-constrained scenarios are prevalent in large-scale model training. For instance, Stable Diffusion, a prominent generative model, pre-trains on 5 billion image-text pairs, while GPT-3, a language model with 175 billion parameters, relies on a vast 45 TB text corpus gathered from various websites for its training. These large models typically operate under a many-to-one (M2O) data collection approach, preventing attackers from accessing the complete training set.
>
> (2) Applications like malware and deepfake detection involve concept drift, where the statistical properties of the target variable—what the model is predicting—evolve over time. Consequently, the training and test samples exhibit differing distributions, aligning with the framework of data-constrained attack scenarios in backdooring for these types of applications.
>
> As part of our future work, we aim to delve into crafting realistic attack scenarios and devising efficient backdoor attacks specifically tailored for these application domains.

---

### Official Review · Reviewer_kSYS · 2023-10-25

**Soundness:** 3 good
**Presentation:** 3 good
**Contribution:** 3 good
**Rating:** 8
**Confidence:** 2

**Summary:**

This paper proposes a new backdoor attack that performs well in data-constraint conditions that are more akin to real-world scenarios. The attack uses the CLIP model as a feature extractor to diminish the entanglement between benign and poison features. The experiment results show significant improvement compared to previous methods in these more realistic conductions.

**Strengths:**

- A novel approach to backdoor attack
- Comprehensive evaluations

**Weaknesses:**

- CLIP limits the application domain
- Defense discussion missing
- Runtime information missing

**Questions:**

The authors present a novel backdoor attack that utilizes the pre-trained CLIP model as a feature extractor to suppress benign features and accentuate poison features. The attack also relaxes previous assumptions that having knowledge of the training datasets and the target models trained on datasets from one distribution. The authors show previous methods do not perform well in these more realistic scenarios but their new method is consistently effective and the trigger is hard to detect visually. Overall, the paper is well-written and the evaluation is comprehensive. However, there are a few points I would like to see the authors to further address.

- The usage of the CLIP model for backdoor attacks is indeed novel. However, this also limits the domains of possible application of the attack. While the method seems to perform well on datasets with natural sceneries, such as CIFAR-100, CIFAR-10, and ImageNet-50, the performance cannot be guaranteed on datasets where the domain drastically differs from CLIP’s training set, such as medical scans, satellite imageries, etc. Additionally, even for similar domains, it would be interesting to see if the feature extraction capabilities transfer onto fine-grained datasets, such as CUB-200-2011, Stanford-Cars, Oxford-Flowers, etc. The authors should consider including results on more diverse datasets.

- The target models used in this paper are all relatively simple/small (experimental settings focused). They also differ drastically from the CLIP model both in terms of architecture and performance. The authors have already pointed out the effect of model architecture in Section 5.1. Evaluating the attack on more advanced and larger architectures, such as ViT, can further prove the author’s claim for applicability in real-world scenarios.

- Discussion regarding potential defenses is also missing. It would be interesting to see how this new attack performs against backdoor detection or defense methods. Since the optimization suppresses the clean features and augments the poison features, defense/detection methods that rely on optimization, such as Neural Cleanse[1] could potentially be more effective (compared to defending against traditional backdoor attacks). Furthermore, a recent work[2] on backdoor defense seems to use similar intuition (detangling benign and poison features). It would be interesting to see how this defense performs against an attack that is intuitively similar.
[1]Wang et al. Neural cleanse: Identifying and mitigating backdoor attacks in neural networks. 2019. In IEEE Symposium on Security and Privacy (S&P).
[2]Min et al. Towards Stable Backdoor Purification through Feature Shift Tuning. 2023. arXiv preprint arXiv:2310.01875.

- Considering the optimization process needed to conduct this attack, the authors should consider including relevant runtime information. Since the focus of this paper is on presenting a backdoor attack that is applicable in real-world scenarios, the computing resource required can be another limiting factor.

Minors:

- Fonts in figures are too small to be legible
- Page 8, VGG-16 datasets? (should be models)

---

> ### Author Response · Authors · 2023-11-16
> **Response for Reviewer kSYS [Cons1-2]**
>
> Thanks for rating our work as novel and comprehensive. To address your concerns, we provide point-wise responses as below.
>
> **Cons1.  The usage of the CLIP model for backdoor attacks is indeed novel. However, this also limits the domains of possible application of the attack.**
>
> **Cons1. Q1. Experiments on remote sensing datasets**
>
> Thank you for your insightful comments, which align with our concerns about the application scope of our approach discussed in Sec. A.12 of the Appendix. The reliance of our technologies on the CLIP model, specifically pre-trained on natural images, may limit their usability in domains like medical imaging or remote sensing. To address this limitation, we propose substituting CLIP with domain-specific pre-trained models, like MedCLIP for medical images or Satellite for remote sensing, tailoring our approach to the target domain.
>
> To delve into the performance in domains distinct from CLIP's training, we conducted experiments on the UCM dataset (Sec. A.13.2 and Table 6). This dataset encompasses 2100 images across 21 categories in remote sensing. The results indicate a performance decline on the UCM dataset. However, as seen in Sec. A.11.1, by adjusting the constraints ($\epsilon$) of the optimized noise, we observed an improved attack success rate in our methods compared to baseline methods. Additionally, replacing CLIP with the Satellite model, a large model fine-tuned on remote sensing images, further augmented the attack success rate. These outcomes demonstrate the adaptability of our methods to various domains by replacing CLIP with domain-specific pre-trained models.
>
>
> **Cons1. Q2. Experiments on fine-grained datasets**
>
> Thanks for your constructive comments. In Sec. A.13.3, we rigorously assess the effectiveness of our proposed methodologies using fine-grained datasets, with a specific focus on the widely recognized Oxford-Flowers dataset within the domain of fine-grained image classification. For our experiments, we set the attack-target class ($k$) to 0 and maintain a consistent poisoning rate of 0.05 (p=307) across all conducted trials. Illustrated in Figure 25 (a), we showcase the Attack Success Rate of number-Constrained backdoor attacks on the Oxford-Flowers dataset. Our findings conclusively demonstrate the superior efficacy of CLIP-based poisoning feature augmentation compared to prior attack methodologies. Additionally, CLIP-based Clean Feature Suppression emerges as a valuable strategy across diverse attack methods. Moreover, as depicted in Figure 25 (b), our results unequivocally indicate that our technologies maintain benign accuracy at par with baseline methods, even under varying settings and diverse backdoor attacks. This underlines the robustness and non-disruptive nature of our methodologies.
>
> **Cons2. Evaluating the attack on more advanced and larger architectures**
>
> Thanks for your constructive comments. In Sec. A.13.4, we assess the effectiveness of our proposed methodologies on more advanced and larger architectures, such as ViT-Small. For our experiments, we set the attack-target class ($k$) to 0 and maintain a consistent poisoning rate of 0.05 (p=500) across all conducted trials. Illustrated in Figure 26 (a), we showcase the Attack Success Rate of number-Constrained backdoor attacks on the ViT-Small architecture and CIFAR-10  dataset. Our findings conclusively demonstrate the superior efficacy of CLIP-based poisoning feature augmentation compared to prior attack methodologies. Additionally, CLIP-based Clean Feature Suppression emerges as a valuable strategy across diverse attack methods. Moreover, as depicted in Figure 26 (b), our results unequivocally indicate that our technologies maintain benign accuracy at par with baseline methods, even under varying settings and diverse backdoor attacks.

---

> ### Author Response · Authors · 2023-11-16
> **Response for Reviewer kSYS [Cons 3-5]**
>
> **Cons3. Discussion regarding potential defenses is also missing.**
>
> Attack stealthiness needs to be evaluated through algorithm. As depicted in Figure 9, our preliminary evaluation has conducted on the VGG-16 and CIFAR-100 datasets vividly demonstrates that our proposed method can attack against pruning-based [1] defense method more effectively than other attack methods.
>
> In addition, our paper includes further results on two other defense methods [2][3] in Section A.13.5. Across these results, a consistent trend emerges—our approach demonstrates a notably higher effectiveness in circumventing defense mechanisms compared to other conventional attack methods. Notably, when tested against the FST defense [3], our proposed methodologies exhibit remarkable enhancements over baseline approaches. A key factor contributing to this success lies in our CLIP-CFE approach, which effectively suppresses the expression of clean features. This suppression prompts fine-tuned based defense methods to excessively focus on learning these clean features, inadvertently suppressing the erasure of poisoning features, thereby amplifying our attack's efficacy.
>
> [1] Fine-pruning: Defending against backdooring attacks on deep neural networks, 2018. In International symposium on research in attacks, intrusions, and defenses.
>
> [2] Wang et al. Neural cleanse: Identifying and mitigating backdoor attacks in neural networks. 2019. In IEEE Symposium on Security and Privacy (S\&P).
>
> [3] Min et al. Towards Stable Backdoor Purification through Feature Shift Tuning. 2023. arXiv preprint arXiv:2310.01875.
>
> **Cons4. The computational overhead and time required for the CLIP optimization process is not extensively analyzed.**
>
> Thanks for your comments. We have included the time overhead of different methods in  the Sec A.11.3 of the Appendix.
>
> We would like to highlight that in the case of BadNet and Blended, adding a pre-defined trigger into benign images to build poisoned images only needs to perform one time addition calculation, which incurs negligible time consumption. It's important to note, however, that these straightforward implementations fall short in terms of both poison efficiency and stealthiness in the context of backdoor attacks. Recent advancements in backdoor attack techniques have sought to enhance efficiency and covert effectiveness by introducing optimization-driven processes to define triggers. While these refinements do entail some additional time overhead, they significantly elevate the attack's efficacy. We undertook an evaluation of the time overhead associated with diverse attack methods using an A100 GPU. As outlined in Table 5, the time consumption exhibited by optimization-based methods grows linearly with the expansion of the poison sample count. Despite this, the overall time overhead remains remarkably modest, with the entirety of the process being accomplished within mere minutes.
>
> **Table 5: The time overhead (min) of different attacks on a A100 GPU. All results are computed with the mean of 5 different runs.**
> | Number of Poisoned set | BadNet | BadNet (CLIP-CFE) |UAP|UAP (CLIP-CFE)|CLIP-UAP|CLIP-UAP (CLIP-CFE)|CLIP-CFA|CLIP-CFA (CLIP-CFE)|
>  | :-----:| :----: | :----: | :----: | :----: | :----: |:----: |:----: |:----: |
> |  500|0 |1| 0.7| 1.6 |1 |2.1| 0.9| 1.9|
>  | 1000 |0| 1.9| 1.5| 3.3| 2| 4| 1.9| 3.9|
> | 2000 |0 |3.7 |2.9| 6.5| 4 |7.9| 3.9 |7.8|
>
> **Cons5. Minors.**
>
> Thank you for your feedback. We apologize for the font size issue in the figures, which might affect legibility. Redrawing all the images is a time-consuming process, significantly impacting our capacity to provide comprehensive and detailed responses during this phase. To ensure a thorough and reliable rebuttal, we kindly request to perform the redraw after the rebuttal phase concludes. This will allow us to dedicate our focus to addressing essential queries more effectively at this stage.

---

### Official Review · Reviewer_nLdg · 2023-10-30

**Soundness:** 2 fair
**Presentation:** 3 good
**Contribution:** 1 poor
**Rating:** 3
**Confidence:** 5

**Summary:**

This paper assumed a threat model for backdoor attacks, so-called as ‘data-constrained backdoor attacks’, where the attacker doesn’t have access to the entire training dataset. Then, the authors claimed that the exiting backdoor attacks are inefficient in this new threat model.

**Strengths:**

The authors considered an interesting topic on AI security, specifically, how to improve the backdoor efficiency in a data-constrained scenario.

**Weaknesses:**

First, the authors only provided the empirical results to support the performance decline when the exiting backdoor attack in the new threat model, as shown in Fig.2. I highly recommend that the authors give a possible theoretical analysis to this phenomenon.

Secondly, the new proposed 'clip-guided backdoor attack' method includes two components: clean feature suppression and poisoning feature augmentation. Specifically, the main idea is to exploit adversarial example to generate the noise to suppress the clean feature or amplify the poison feature. Unfortunately, as far as I know this idea has been exploited by many published papers, for instance, as shown as follows. The main difference of this paper is that it is based on a novel pre-trained model CLIP.

[1] Zhao, Shihao, et al. "Clean-label backdoor attacks on video recognition models." Proceedings of the IEEE/CVF conference on computer vision and pattern recognition. 2020.
[2] Turner, D. Tsipras, and A. Madry, “Label-consistent backdoor attacks,” arXiv preprint arXiv:1912.02771, 2019.

In summary, the main idea has been exploited already, which will significantly reduce the contribution of this paper.

**Questions:**

What is the main difference between the 'clip-guided backdoor attack' with the existing references which have been mentioned in the 'weaknesses'

---

> ### Author Response · Authors · 2023-11-12
> **Response for Reviewer nLdg [Cons1]**
>
> **Cons1. Lack of theoretical justification.**
>
> Thank you for your insightful feedback. In considering the rationale for shortcut learning in neural networks, [1] offers both theoretical and empirical evidence indicating that neural networks are inherently inclined to learn from the entirety of input features. Consequently, neural networks exhibit a certain predisposition toward utilizing all available features comprehensively in their decision-making processes. This tendency leads to the activation of both poisoning and benign features during backdoor injection, a crucial insight crucial to understanding the observed behavior.
>
> Additionally, within Section A.3 of the Appendix, we present **three additional observations** regarding data-constrained backdoor attacks. These observations serve to highlight the intrinsic entanglement between benign and poisoning features during the backdoor injection process. This entanglement stands as a principal contributing factor to the failure of current attack methods in data-constrained scenarios. These insights shed light on the challenges inherent in such scenarios and contribute significantly to our understanding of the limitations of existing attack methodologies.
>
> **Observation 1: BadNet outperforms Blended notably in data-constrained attack scenarios.** In a practical experimentation setting, BadNet and Blended exhibit comparable performance under unrestricted attack conditions. Conversely, in data-constrained attack scenarios, BadNet outperforms Blended notably. This intriguing disparity requires elucidation. BadNet employs a $2 \times 2$ attacker-specified pixel patch as a universal trigger pattern attached to benign samples, whereas Blended employs an attacker-specified benign image for the same purpose. Comparatively, Blended's trigger exhibits greater feature similarity to benign images, engendering a more pronounced feature entanglement between poisoned and benign attributes. Accordingly, the performance of the blended dirty-label single-class scenario significantly lags behind other cases, lending credence to our hypothesis that entanglement underpins the degradation of data-constrained backdoor attacks.
>
> **Observation 2: Attack efficiency of number-constrained, dirty-label single-class, and cleanlabel single-class backdoor attacks decreases in turn under the same poison rate.**
> To understand the reason behind this phenomenon, we provide visualizations of the backdoor injection and activation phases for these three attacks in Fig. 6 (b). For the number-constrained backdoor attack, the distribution of poisoning samples in the backdoor injection phase is the same as that in the backdoor activation phase. In other words, both benign and poisoning features are activated simultaneously during both phases. However, for the dirty-label single-class backdoor attack, the distribution of poisoning samples (consisting of single-class benign and poisoning features) in the backdoor injection phase is different from that in the backdoor activation phase. During the injection phase, both benign and poisoning features are activated, but during activation phase, only the poisoning feature is activated. This is the reason why previous attack methods on dirty-label single-class backdoor attacks exhibit performance degeneration. The clean-label single-class backdoor attack is similar to the dirty-label single-class backdoor attack in terms of the distribution of poisoning samples. However, during backdoor injection, there is competing activation between benign and poisoning features. Consequently, the poisoning efficiency of clean-label single-class backdoor attacks is lower than that of dirty-label single-class backdoor attacks.
>
> **Observation 3: Substantial dissimilarities in activation between poisoned samples that share the same trigger.** We have embraced Grad-CAM to more effectively corroborate the presence of a correlation between the benign and the backdoor features. To substantiate this hypothesis, we have incorporated Grad-CAM outcomes to our study. Specifically, we have applied this technique to the BadNet-based poison model on the CIFAR-100 dataset, as depicted in Figure 7. The discernible results from these Grad-CAM visualizations underscore the substantial dissimilarities in activation between poisoned samples that share the same trigger. This visual evidence compellingly demonstrates the intricate entanglement existing between the benign and the backdoor features during instances of backdoor attacks.
>
> For an in-depth exploration, we invite readers to refer to Section A.3 in the Appendix. We trust that this detailed analysis will effectively address the reviewer's concerns regarding the theoretical aspects of our approach.
>
> [1] Shortcut learning in deep neural networks. Nature Machine Intelligence.

---

> ### Author Response · Authors · 2023-11-12
> **Response for Reviewer nLdg [Cons2]**
>
> **Cons2. Lack of novelty**
>
> We respectfully argue our conclusions are novel and fundamentally different from existing literature.
>
> 1. Our setting and problems are novel, which is the main contribution of our study.
>
> (1) First, in this paper, we present **a novel and contemporary backdoor attack scenario** called data-constrained backdoor attacks, which assumes that attackers lack access to the entire training data, making it a versatile and practical attack with broad applicability. References [1][2] design adversarial example as trigger through using the proxy model that is pre-trained on the entire training set, which is not accessible in our settings.
>
> (2) Second, we present three observations pertaining to data-constrained backdoor attacks, serving as compelling evidence for the presence of a significant interdependence between benign and poisoning features during the backdoor injection. This intricate entanglement is identified as the primary factor responsible for the inadequacies exhibited by current attack methodologies within data-constrained scenarios. Our study is a **pioneering exploration of feature entanglement in the context of backdoor attacks**, yielding fresh and valuable insights into the realm of backdoor learning. Ideally, one would anticipate backdoor models to exclusively rely on poisoning features when confronted with poisoning samples, as this would be the most efficient strategy for executing successful backdoor attacks. However, neural networks tend to be greedy and utilize all features for decision-making, leading to activation of both poisoning and benign features during backdoor injection. This results in reduced poisoning efficiency when there is a difference in benign features between the backdoor injection and activation phases, as evidenced in data-constrained backdoor attacks.
>
> 2. Our proposed methods offer a novel approach in the domain of backdoor attacks.
>
> (1) First, while prior studies have focused solely on poisoning feature augmentation, our work introduces a pioneering concept of clean feature suppression. To the best of our knowledge, **no existing research has previously considered this dual strategy**, marking a significant departure in the exploration of backdoor attacks. This novel perspective not only presents a fresh insight but also expands the available options in the realm of backdoor attack methodologies.
>
> (2) Second, our introduction of a CLIP-based clean feature erasing method represents the **inaugural application of machine unlearning in the context of backdoor attacks**.
>
> (3) Third, our proposed CLIP-based contrastive feature augmentation (CLIP-CFA) stands out as a versatile trigger design independent of the attack-target label. Drawing from the entanglement between benign and poisoning features, we employ contrastive optimization to enhance poisoning feature augmentation. This CLIP-CFA approach aims to render the poisoning feature extracted from the designed trigger more expressive compared to the clean feature from the clean samples. Notably, this marks **the first instance of applying contrastive learning in backdoor trigger design**.
>
> (4) Fourth, while our work draws inspiration from previously published papers [1][2] regarding CLIP-based universal adversarial perturbations, it diverges in terms of the optimization objective ([1][2] optimize the perturbation through minimizing cross entropy loss, while our CLIP-UAP optimizes the perturbation through minimizing the CLIP's zero-shot classification loss). Furthermore, our study marks the **initial exploration of the utilization of the CLIP model in the domain of backdoor attacks**, underscoring its pioneering nature in this field.
>
> [1] Clean-label backdoor attacks on video recognition models. CVPR 2020
>
> [2] Label-consistent backdoor attacks. Arxiv 2019.
>
> Our study represents a novel contribution at both the problem and methodological levels. We aim for our detailed explanation to effectively address any concerns regarding the uniqueness of our work. Your feedback is invaluable to us; please feel free to share any additional comments or suggestions. Thank you once again for your valuable insights!

---

> ### Author Response · Authors · 2023-11-20
> **Sincerely expecting further discussions from Reviewer nLdg**
>
> Dear Reviewer nLdg,
>
> We thank reviewer nLdg time for the review and constructive comments. We really hope to have a further discussion with reviewer nLdg to see if our response solves the concerns.
>
> We would sincerely appreciate it if reviewer nLdg could reply to the most important points in our rebuttal. As we pointed out, our study contsins various novelty in terms of attack scenario and methodology. Specifically,
>
> (1)  We present a novel yet real-world backdoor attack scenario, data-constrained backdoor attacks.
>
> (2) Our study is a pioneering exploration of entanglement between benign and poisoning faetures in the context of backdoor attacks, yielding fresh and valuable insights into the realm of backdoor learning.
>
> (3) To the best of our knowledge, no existing research has previously considered both poisoning feature augmentation and clean feature suppression, marking a significant departure in the exploration of backdoor attacks.
>
> (4) Our introduction of a CLIP-based clean feature erasing method represents the inaugural application of machine unlearning in the context of backdoor attacks.
>
> (5) Our study is the first instance of applying contrastive learning in backdoor trigger design.
>
> (6) Our study marks the initial exploration of the utilization of the CLIP model in the domain of backdoor attacks.
>
> We genuinely hope reviewer nLdg could kindly check our response. Thank you!
>
> Best wishes,
>
> Authors

---

> ### Author Response · Authors · 2023-11-22
> **Sincerely expecting further discussions from Reviewer nLdg**
>
> Dear Reviewer nLdg,
>
> We really appreciated your time and constructive reviews. We politely send you a kind reminder that the discussion period is ending within 30 hrs.
>
> Currently, the other three reviewers had acknowledged the contributions and novelty of this work, and they had made a consensus of the acceptance. Given our detailed replies and new experiments, could you please kindly check our response to see if it solves your concerns, so that you could also support our acceptance? Your support is very important to us and we greatly appreciate that!
>
> Meantime, please do not hesitate to reach out to us if there are other clarifications or experiments we can provide. Many thanks!
>
> Best Regards,
>
> Authors

---

### Official Review · Reviewer_rJBe · 2023-11-01

**Soundness:** 3 good
**Presentation:** 2 fair
**Contribution:** 2 fair
**Rating:** 6
**Confidence:** 4

**Summary:**

This paper addresses an important and practical backdoor attack scenario called data-constrained backdoor attacks. The key insight is that in real-world settings, attackers often do not have full access to a victim's entire training dataset, which spans multiple sources. The paper clearly defines three variants of data-constrained attacks based on restrictions on the number of poisoning samples, classes, or domains.
A thorough set of experiments on CIFAR and ImageNet datasets demonstrates that existing backdoor methods like BadNets and Blended attacks fail under data constraints, due to entanglement between benign and poisoning features. The analysis of this entanglement issue is a nice contribution. To address this limitation, the authors cleverly utilize CLIP in two ways: 1. Clean feature suppression via CLIP-CFE to erase benign features.
2. Poisoning feature augmentation via CLIP-UAP and CLIP-CFA to amplify poisoning features.
The introduction of CLIP for backdoor attacks is novel. Results show CLIP-UAP and CLIP-CFA consistently outperform baseline triggers across constraints, architectures, and datasets. CLIP-CFE provides further improvements in attack success rate. The attacks remain stealthy and do not impact benign accuracy.

**Strengths:**

1.	Addresses a highly practical attack scenario of data-constrained backdoor attacks that reflects real-world training environments where attackers have limited data control.
2.	Provides a clear taxonomy of data-constrained attacks based on restrictions to number of samples, classes, and domains.
3.	Identifies through analysis and experiments that existing attacks fail under data constraints due to entanglement of benign and poisoning features. This is an important insight.

**Weaknesses:**

1.	While the data-constrained scenario is practical, the specific sub-variants of number, class, and domain constraints may not fully capture all real-world limitations an attacker could face. More complex constraints could be studied.
2.	The computational overhead and time required for the CLIP optimization process is not extensively analyzed. This could be a limitation for realistic attacks.
3.	The stealthiness metrics mainly rely on signal processing based measures like PSNR and SSIM. More rigorous stealthiness analysis like visualizations and defense evaluations may be beneficial.

**Questions:**

see in weakness

---

> ### Author Response · Authors · 2023-11-16
> **Response for Reviewer rJBe[Cons1-2]**
>
> Many thanks to Reviewer rJBe for acknowledging our work as valuable and convincing. To address your concerns, we provide point-wise answers and extra experiment results as below.
>
> **Cons1. More complex constraints could be studied.**
>
> Figure 4 in the manuscript showcases a series of ablation studies that demonstrate the performance of various attacks under different sub-variants of number, class, and domain constraints. Notably, we implemented the most stringent settings: setting the poison data at 0.004, poison classes at 1, and domain rate at 0 for the number-constraints, class-constraints, and domain-constraints backdoor attacks, respectively. The results demonstrate that our proposed methods still be viable in  most stringent scenarios.
>
> Furthermore, in Section A.13.1 of the manuscript, we extend our exploration to encompass more intricate constraints within data-constraint backdoor attacks. Specifically, we investigate two additional configurations:
>
> **Config A**: poisoning rate =  0.01 (P=500), poisoning classes=1 (1), and domain rate=0.5.
>
> **Config B**: poisoning rate =  0.01 (P=500), poisoning classes=3, and domain rate=0.25.
>
> The experiments are performed using the VGG-16 model and the CIFAR-100 dataset. As depicted in Fig. 24, our methods and conclusions are demonstrated to be equally viable in scenarios involving more complex data constraints.
>
> **Cons2. The computational overhead and time required for the CLIP optimization process is not extensively analyzed.**
>
> Thanks for your comments. We have included the time overhead of different methods in  the Sec A.11.3 of the Appendix.
>
> We would like to highlight that in the case of BadNet and Blended, adding a pre-defined trigger into benign images to build poisoned images only needs to perform one time addition calculation, which incurs negligible time consumption. It's important to note, however, that these straightforward implementations fall short in terms of both poison efficiency and stealthiness in the context of backdoor attacks. Recent advancements in backdoor attack techniques have sought to enhance efficiency and covert effectiveness by introducing optimization-driven processes to define triggers. While these refinements do entail some additional time overhead, they significantly elevate the attack's efficacy. We undertook an evaluation of the time overhead associated with diverse attack methods using an A100 GPU. As outlined in Table 5, the time consumption exhibited by optimization-based methods grows linearly with the expansion of the poison sample count. Despite this, the overall time overhead remains remarkably modest, with the entirety of the process being accomplished within mere minutes.
>
> **Table 5: The time overhead (min) of different attacks on a A100 GPU. All results are computed with the mean of 5 different runs.**
> | Number of Poisoned set | BadNet | BadNet (CLIP-CFE) |UAP|UAP (CLIP-CFE)|CLIP-UAP|CLIP-UAP (CLIP-CFE)|CLIP-CFA|CLIP-CFA (CLIP-CFE)|
>  | :-----:| :----: | :----: | :----: | :----: | :----: |:----: |:----: |:----: |
> |  500|0 |1| 0.7| 1.6 |1 |2.1| 0.9| 1.9|
>  | 1000 |0| 1.9| 1.5| 3.3| 2| 4| 1.9| 3.9|
> | 2000 |0 |3.7 |2.9| 6.5| 4 |7.9| 3.9 |7.8|

---

> ### Author Response · Authors · 2023-11-16
> **Response for Reviewer rJBe[Cons 3]**
>
> **Cons3. More rigorous stealthiness analysis like visualizations and defense evaluations may be beneficial.**
>
> We've undertaken a more comprehensive analysis of stealthiness, involving visualizations and defense evaluations in Section A.8 of the Appendix.
>
> **visualizations:** In Fig. 8, we present examples of poisoning images generated by various attacks on the ImageNet-50 dataset. Given the low resolution of CIFAR-10 and CIFAR-100, clear visualizations are challenging. Consequently, visualizations are exclusively provided for the ImageNet-50 dataset. The results depict the visual invisibility of our proposed methods—CLIP-UAP, CLIP-CFA, and CLIP-CFE—in comparison to BadNets and Blended attacks.
>
> **Stealth in attack defense methods:** Attack stealthiness needs to be evaluated through algorithm. As depicted in Figure 9, our preliminary evaluation conducted on the VGG-16 and CIFAR-100 datasets vividly demonstrates that our proposed method can attack against pruning-based [1] defense method more effectively than other attack methods.
>
> In addition, our paper includes further results on two other defense methods [2][3] in Section A.13.5. Across these results, a consistent trend emerges—our approach demonstrates a notably higher effectiveness in circumventing defense mechanisms compared to other conventional attack methods. Notably, when tested against the FST defense [3], our proposed methodologies exhibit remarkable enhancements over baseline approaches. A key factor contributing to this success lies in our CLIP-CFE approach, which effectively suppresses the expression of clean features. This suppression prompts fine-tuned based defense methods to excessively focus on learning these clean features, inadvertently suppressing the erasure of poisoning features, thereby amplifying our attack's efficacy.
>
> [1] Fine-pruning: Defending against backdooring attacks on deep neural networks, 2018. In International symposium on research in attacks, intrusions, and defenses.
>
> [2] Wang et al. Neural cleanse: Identifying and mitigating backdoor attacks in neural networks. 2019. In IEEE Symposium on Security and Privacy (S\&P).
>
> [3] Min et al. Towards Stable Backdoor Purification through Feature Shift Tuning. 2023. arXiv preprint arXiv:2310.01875.

---

### Author Response · Authors · 2023-11-16
**General response**

We extend our gratitude to all the reviewers, PCs, and ACs for their invaluable efforts. Below, we present our point-to-point responses. Additionally, we have included Section A.13 in the manuscript's appendix, comprising detailed experiments. These encompass exploring more intricate constraints within data-constraint backdoor attacks (Section A.13.1), experimenting with domains distinct from CLIP's training domain (Section A.13.2), analyzing on fine-grained datasets (Section A.13.3), experiments on the ViT architecture (Section A.13.4), discussing potential defenses (Section A.13.5), and delineating the distinction from recent state-of-the-art attacks (Section A.13.6).  We hope our efforts can effectively address any concerns regarding the weakness of our work. We welcome any further comments or suggestions. Thank you for your valuable insights!

---

### Author Response · Authors · 2023-11-20
**Sincerely expecting further discussions from Reviewers**

Dear Reviewers,

We sincerely hope to have further discussion with reviewers to see if our response solves your concerns. We are confident that our response should have cleared the air, and we can clarify more if there is more need. We are happy to answer any additional questions and provide more information.

We genuinely hope reviewers could kindly check our response. Thank you!

Best wishes,

Authors

---

### Meta-Review · Area_Chair_THxa · 2023-12-12

**Metareview:**

The rebuttal addressed some of the concerns. Reviewer nLdg is the only one not recommending acceptance. However, I believe their concerns are addressed in the rebuttal and they did not comment after the rebuttal. For instance I do not think the theoretical analysis suggested by Reviewer nLdg is really needed for this submission. Hence, since other reviewers are positive, the paper will be accepted.

The submission still has some weaknesses including Insufficient experimental results and analysis of the computational overhead. The main weaknesses are:
- The analysis of the computational overhead and time needed for the CLIP optimization process is not thorough.
- The metrics assessing stealthiness primarily depend on signal processing measures such as PSNR and SSIM. A more comprehensive stealthiness analysis, including visualizations and defense evaluations, could be advantageous.
- Missing sufficient run-time analysis and defense discussion.

**Justification For Why Not Higher Score:**

The submission still has some weaknesses including Insufficient experimental results and analysis of the computational overhead. The main weaknesses are:
- The analysis of the computational overhead and time needed for the CLIP optimization process is not thorough.
- The metrics assessing stealthiness primarily depend on signal processing measures such as PSNR and SSIM. A more comprehensive stealthiness analysis, including visualizations and defense evaluations, could be advantageous.
- Missing sufficient run-time analysis and defense discussion.

**Justification For Why Not Lower Score:**

The submission has novelty and is relatively more practical compared to many other works.

---

### Decision · Program_Chairs · 2024-01-16

Accept (poster)